# Learning interpretable cellular and gene signature embeddings from single-cell transcriptomic data

Yifan Zhao [1,5,6], Huiyu Cai [2,6], Zuobai Zhang[3], Jian Tang[4✉] & Yue Li [1✉]

The advent of single-cell RNA sequencing (scRNA-seq) technologies has revolutionized transcriptomic studies. However, large-scale integrative analysis of scRNA-seq data remains a challenge largely due to unwanted batch effects and the limited transferabilty, interpretability, and scalability of the existing computational methods. We present single-cell Embedded Topic Model (scETM). Our key contribution is the utilization of a transferable neural-network-based encoder while having an interpretable linear decoder via a matrix tri-factorization. In particular, scETM simultaneously learns an encoder network to infer cell type mixture and a set of highly interpretable gene embeddings, topic embeddings, and batch-effect linear intercepts from multiple scRNA-seq datasets. scETM is scalable to over $10^6$ cells and confers remarkable cross-tissue and cross-species zero-shot transfer-learning performance. Using gene set enrichment analysis, we find that scETM-learned topics are enriched in biologically meaningful and disease-related pathways. Lastly, scETM enables the incorporation of known gene sets into the gene embeddings, thereby directly learning the associations between pathways and topics via the topic embeddings.

[1] School of Computer Science, McGill University, Montreal, QC, Canada. [2] Department of Machine Intelligence, Peking University, Beijing, China. [3] School of Computer Science, Fudan University, Shanghai, China. [4] HEC Montreal, Montreal, QC, Canada. [5]Present address: Harvard-MIT Health Sciences and Technology, Cambridge, MA, USA. [6]These authors contributed equally: Yifan Zhao, Huiyu Cai. ✉email: jian.tang@hec.ca; yueli@cs.mcgill.ca

Advances in high-throughput sequencing technologies[1] provide an unprecedented opportunity to profile the individual cells' transcriptomes across various biological and pathological conditions, and have spurred the creation of several atlas projects[2–5]. Emerged as a key application of single-cell RNA sequencing (scRNA-seq) data, unsupervised clustering allows for cell-type identification in a data-driven manner. Flexible, scalable, and interpretable computational methods are crucial for unleashing the full potential from the wealth of single-cell datasets and translating the transcription profiles into biological insights. Despite considerable progress made on clustering method development for scRNA-seq data analysis[6–16], several challenges remain.

First, compared to bulk RNA-seq, scRNA-seq data commonly exhibit higher noise levels and drop-out rates, where the data only captures a small fraction of a cell's transcriptome[17]. Changes in gene expression due to experimental design, often referred to as batch effects[18], can have a large impact on clustering[12,18–20]. If not properly addressed, these technical artefacts may mask the true biological signals in cell clustering.

Second, the partitioning of the cell population alone is insufficient to produce biological interpretation. The annotations of the cell clusters require extensive manual literature search in practice and the annotation quality may be dependent on users' domain knowledge[20]. Therefore, an interpretable and flexible model is needed. In the current work, we consider model interpretability as whether the model parameters can be directly used to associate the input features with latent factors or target outcomes. Latent topic models are a popular approach in mining genomic and healthcare data[21–23] and are increasingly being used in the scRNA-seq literature[24]. Specifically, in topic modeling, we infer the topic distribution for both the samples and genomic features by decomposing the samples-by-features matrix into samples-by-topics and topics-by-features matrices, which also be viewed as a probabilistic non-negative factorization (NMF)[25]. Importantly, the top genes under each latent topic can reveal the gene signatures for specific cellular programs, which may be shared across cell types or exclusive to a particular cell type. Traditionally, the latter are detected via differential expression analysis at individual gene levels, which has limited statistical power in scRNA-seq data analysis because of the sparse gene counts, small number of unique biological samples, and the burdens of multiple testings.

Third, model transferrability is an important consideration. We consider a model as transferable if the learned knowledge manifested as the model parameters could benefit future data modeling. In the context of scRNA-seq data analysis, it translates to learning feature representations from one or more large-scale reference datasets and applying the learned representations to a target dataset[26,27]. If the model is not further trained on the target dataset, the learning setting called zero-shot transfer learning. A model that can successfully separate cells of distinct cell types that are not present in the reference dataset implies that the model has learned some meaningful abstraction of the cellular programs from the reference dataset such that it can generalize to annotating new cell types of different kinds. An analogy would be that someone who has learned how to distinguish triangles from rectangles may also be able to distinguish squares from circles. As the number and size of scRNA-seq datasets continue to increase, there is an increasingly high demand for efficient exploitation and knowledge transfer from the existing reference datasets.

Several recent methods have attempted to address these challenges. Seurat[7] uses canonical correlation analysis to project cells onto a common embedding, then identifies, filters, scores, and weights anchor cell pairs between batches to perform data integration. Harmony[28] iterates between maximum diversity clustering and a linear batch correction based on the mixture-of-experts model. Scanorama[10] performs all-to-all dataset matching by querying nearest neighbors of a cell among all remaining batches, after which it merges the batches with a Gaussian kernel to form a single-cell panorama. These methods rely on feature (gene) selection and/or dimensionality reduction methods; otherwise they can not scale to compendium-scale reference[2] or cohort-level single-cell transcriptome data[29] or are sensitive to the noise inherent to scRNA-seq count data. They are also non-transferable, meaning that the knowledge learned from one dataset cannot be easily transferred through model parameter sharing to benefit the modeling of another dataset. NMF approaches such as UNCURL[30] works only with one scRNA-seq dataset. LIGER[9] uses integrative NMF to jointly factorize multiple scRNA-seq matrices across conditions using genes as the common axis, linking cells from different conditions by a common set of latent factors also known as metagenes. LIGER is weakly transferable in the sense that the global metagenes-by-genes matrix can be recycled as initial parameters when modeling a new target dataset, whereas the cells-by-metagenes and the final metagenes-by-genes must be further computed and updated by iterative numerical optimization to fit the target dataset.

Deep-learning approaches, especially autoencoders, have demonstrated promising performance in scRNA-seq data modeling. scAlign[15] and MARS[31] encode cells with non-linear embeddings using autoencoders, which is naturally transferable across datasets. While scAlign minimizes the distance between the pairwise cell similarity at the embedding and original space, MARS looks for latent landmarks from known cell types to infer cell types of unknown cells. Variational autoencoders (VAE)[32] is an efficient Bayesian framework for approximating intractable posterior distribution using proposed distribution parameterized by neural networks. Several recent studies have tailored the original VAE framework towards modeling single-cell data. Single-cell variational inference (scVI)[6] models library size and takes into account batch effect in generating cell embeddings. scVAE-GM[11] changed the prior distribution of the latent variables in the VAE from Gaussian to Gaussian mixture model, adding a categorical latent variable that clusters cells. Lotfollahi et al.[26] developed a VAE model called scGen to infer the expression difference due to perturbation conditions by latent space interpolation. A key drawback for these VAE models is the lack of interpretability—post hoc analyses are needed to decipher the learned model parameters and distill biological meaning from the learned network parameters. To improve interpretability, Svensson et al.[14] developed a linear decoded VAE (hereafter referred to as scVI-LD) as a part of the scVI software.

In this paper, we present single-cell Embedded Topic Model (scETM), a generative topic model that facilitates integrative analysis of large-scale single-cell transcriptomic data. Our key contribution is the utilization of a transferable neural-network-based encoder while having an interpretable linear decoder via a matrix tri-factorization. The scETM simultaneously learns the encoder network parameters and a set of highly interpretable gene embeddings, topic embeddings, and batch-effect linear intercepts from scRNA-seq data. The flexibility and expressiveness of the encoder network enable scETM to model extremely large scRNA-seq datasets without the need of feature selection or dimension reduction. By tri-factorizing cells-genes matrix into cells-by-topics, topics-by-embeddings, and embeddings-by-genes, we are able to incorporate existing pathway information into gene embeddings during the model training to further improve interpretability. This is a salient feature compared to related methods such as scVI-LD. It allows scETM to simultaneously discover interpretable cellular signatures and gene markers while

(a) scETM modeling of single-cell transcriptomes across multiple experiments or studies

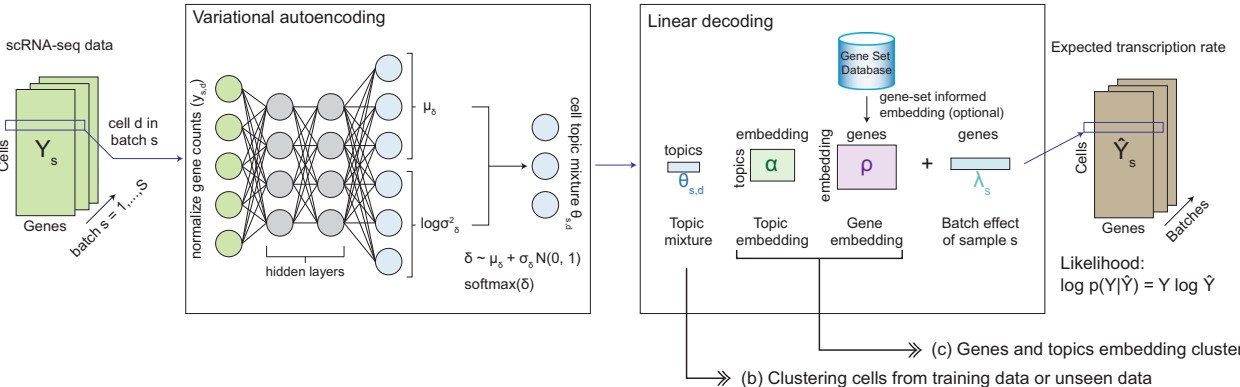

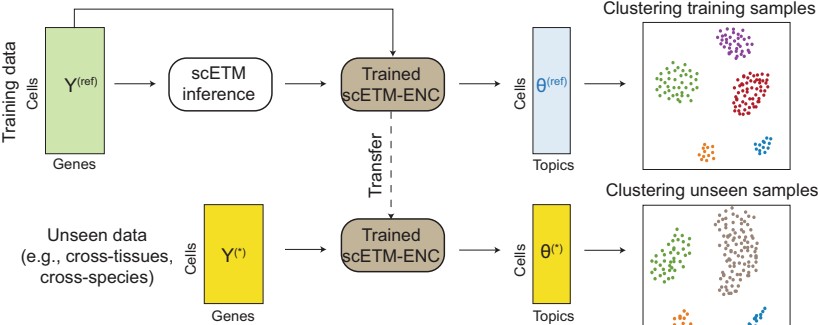

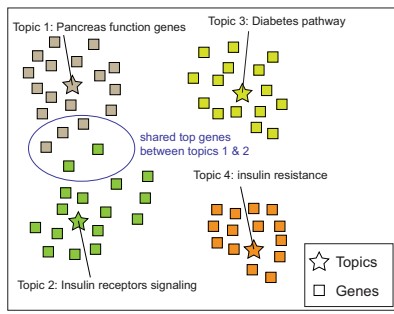

**Fig. 1 scETM model overview. a** scETM training. Given as input the scRNA-seq data matrices across multiple experiments or studies (i.e., batches), scETM models the single-cell transcriptomes using an embedded topic-modeling approach. Each scRNA-seq profile serves as an input to a variational autoencoder (VAE) as the normalized gene counts. The encoder network produces a stochastic sample of the latent topic mixture ($\theta_{s,d}$ for batch $s = 1, ..., S$ and cell $d = 1, ..., N_s$), which can be used for clustering cells (see panel **b**). The linear decoder learns topic embedding and gene embedding, which can be used to analyze cellular programs via enrichment analyses (see panel **c**). **b** Workflow used to perform zero-shot transfer learning. The trained scETM-encoder on a reference scRNA-seq dataset is used to infer the cell topic mixture $\theta^*$ from an unseen scRNA-seq dataset without training them. The resulting cell mixtures are then visualized via UMAP visualization and evaluated by standard unsupervised clustering metrics using the ground-truth cell types. **c** Exploring gene embeddings and topic embeddings. As the genes and topics share the same embedding space, we can explore their connections via UMAP visualization or annotate each topic via enrichment analyses using known pathways.

integrating scRNA-seq data across conditions, subjects and/or experimental studies.

We demonstrate that scETM offers state-of-the-art performance in clustering cells into known cell types across a diverse range of datasets with desirable runtime and memory requirements. We also demonstrate scETM's capability of effective knowledge transfer between different sequencing technologies, between different tissues, and between different species. We then use scETM to discover biologically meaningful gene expression signatures indicative of known cell types and pathophysiological conditions. We analyze scETM-inferred topics and show that several topics are enriched in cell-type-specific or disease-related pathways. Finally, we directly incorporate known pathway-gene relationships (pathway gene set databases) into scETM in the form of gene embeddings, and use the learned pathway-topic embedding to show the pathway-informed scETM (p-scETM)'s capability of learning biologically meaningful information.

## Results

**scETM model overview.** We developed scETM to model scRNA-seq data across experiments or studies, which we term as batches (Fig. 1a and Supplementary Fig. 1a). Adapted from the Embedded Topic Model (ETM)[33], scETM inherits the benefits of topic models, and is effective for handling large and heavy-tailed

distribution of word frequency. In the context of scRNA-seq data analysis, each sampled single-cell transcriptome is provided as a vector of normalized gene counts to a two-layer fully-connected neural-network (i.e., encoder; see detailed architecture in Supplementary Fig. 1c), which infers the topic mixing proportions of the cell. The trained encoder on a reference scRNA-seq data can be used to infer topic mixture of unseen scRNA-seq data collected from different tissues or species (Fig. 1b).

For interpretability, we use a linear decoder with the gene and topic embeddings as the learnable parameters. Specifically, we factorize the cells-by-genes count matrix into a cells-by-topics matrix $\theta$ (inferred by the encoder), topics-by-embedding $\alpha$, and embedding-by-genes $\rho$ matrices (Supplementary Fig. 1b). This tri-factorization design allows for exploring the relations among cells, genes, and topics in a highly interpretable way. To account for biases across conditions or subjects, we introduce an optional batch correction parameter $\lambda$, which acts as a linear intercept term in the categorical softmax function to alleviate the burden of modeling batch effects from the encoder to let it focus on inferring biologically meaningful cell topic mixture $\theta_d$. The encoder and embedding learning is performed by an amortized variational inference algorithm to maximize the evidence lower bound (ELBO) of the marginal categorical likelihood of the scRNA-seq counts[32]. Compared to scVI-LD[14], the linear decoder component that learns a common embeddings for both topics

**Table 1 Model properties and unsupervised clustering performance on data integration tasks.**

|  | Transferable | Interpretable | MP | HP | TM | AD | MDD | MR |
|---|---|---|---|---|---|---|---|---|
| Harmony |  |  | 0.969 | 0.955 | 0.705 | 0.994 | 0.784 | 0.763 |
| Scanorama |  |  | 0.915 | 0.859 | 0.542 | 0.997 | 0.587 | 0.780 |
| Seurat |  |  | 0.944 | 0.968 | 0.676 | 0.991 | 0.550 | 0.781 |
| scVAE-GM | ✓ |  | 0.805 | NA | NA | 0.997 | 0.563 | 0.778 |
| scVI | ✓ |  | 0.932 | 0.759 | 0.670 | 0.991 | 0.541 | 0.783 |
| LIGER | ✓ | ✓ | 0.914 | 0.911 | 0.591 | 0.894 | 0.704 | 0.714 |
| scVI-LD | ✓ | ✓ | 0.875 | 0.656 | 0.608 | 0.989 | 0.666 | 0.718 |
| scETM | ✓ | ✓ | 0.946 | 0.943 | 0.761 | 0.996 | 0.717 | 0.859 |
| scETM $-\lambda$ | ✓ | ✓ | 0.851 | 0.474 | 0.629 | 0.996 | 0.719 | 0.773 |
| scETM $+$ adv | ✓ | ✓ | 0.944 | 0.946 | 0.704 | 0.993 | 0.717 | 0.772 |
| Batch Effect |  |  | Strain | Tech. | Tech. | Ind. | Ind. | Studies |

The clustering performance is measured by adjusted rand index (ARI) between ground-truth cell types and Leiden[75] clusters. scETM performances with or without the linear batch correction (scETM, scETM $-\lambda$) are both reported. scETM $+$ adv is scETM plus adversarial network loss to further correct batch effects. Batch variables include strain, sequencing technologies (Tech.) and individuals (Ind.). NA is reported for models that did not converge. Experimental details are described in the "Methods" section. We ran Seurat on all datasets with integration turned on whenever applicable.

and genes offers more flexibility and interpretability and overall better performance as we demonstrate next (Fig. 1c). Details of the scETM algorithm and implementation are described in Methods.

**Data integration**. We benchmarked scETM, along with seven state-of-the-art single-cell clustering or integrative analysis methods—scVI[6], scVI-LD[14], Seurat v3[7], scVAE-GM[11], Scanorama[10], Harmony[28] and LIGER[9], on six published datasets, namely Mouse Pancreatic Islet (MP)[34], Human Pancreatic Islet (HP)[7], Tabula Muris (TM)[3], Alzheimer's Disease dataset (AD)[35], Major Depressive Disorder dataset (MDD)[29], and Mouse Retina (MR)[36,37] (Supplementary Methods). Across all datasets, scETM stably delivered competitive results especially among the transferable and interpretable models, while others methods fluctuate across different datasets in terms of Adjusted Rand Index (ARI) and Normalized Mutual Information (NMI) (Table 1; Supplementary Table 1). Overall, Harmony and Seurat have slightly higher ARIs than scETM, with trade-offs of model transferrability, interpretability, and/or scalability, which we investigate in the following sections.

We further experimented the same scETM without the batch correction term, namely scETM-$\lambda$. Compared to the $\lambda$-ablated model, the full scETM model confers higher ARI in 3 out of the 5 datasets (Table 1) and higher NMI in 4 out of the 5 datasets (Supplementary Table 1). Improvement over the Human Pancreas (HP) dataset is remarkably high, implying an effective correction of the confounder due to the scRNA-seq technology differences. We observe no improvement in the AD dataset in terms of both ARI and NMI and small improvement in MDD only in terms of NMI. This implies a lesser concern of batch effects from only the individual brain sample donors, with all data being collected by the same technology in a single study.

We also evaluated the batch mixing aspect of scETM and other methods using k-nearest-neighbor Batch-Effect Test (kBET)[18] (Table 2) and examined to what extent scETM's batch mixing performance can be improved by introducing an adversarial loss term to scETM (Methods)[38]. Briefly, we used a discriminator network (a two-layer feed-forward network) to predict batch labels using the cell topic mixture embeddings generated by the encoder network, and directed the encoder network to fool the discriminator. We observe notable improvement on kBET with similar ARI and NMI scores (Table 1 and Supplementary Table 1, row "scETM+adv") at the cost of up to 50% more running time. This shows the expandability of scETM. For the subsequent analyses, we opted to use the results from scETM (without the adversarial loss but with the linear batch correction $\lambda$) because of

its simpler design, scalability, comparable ARI scores, and less aggressive batch correction (see below).

scETM is also robust to architectural and hyperparameter changes, requiring very few or no architecture adaptation or hyperparameter tuning efforts when applied to unseen datasets (Supplementary Table 2). As a result, we used the same architecture and hyperparameters for all datasets in Table 1. We also performed a comprehensive ablation analysis to validate our model choices. The ablation experiment demonstrates the necessity of the key model components, such as the batch-effect correction factors $\lambda$ and the batch normalization technique used in training the encoder. Normalizing gene expression scRNA-seq counts as the input to the encoder also improves the performance (Supplementary Table 3).

Clustering agreement metrics are not the only metrics for evaluating scRNA-seq methods, and are not available to unannotated datasets. Therefore, we also evaluated the negative log-likelihood (NLL) on held-out samples, which is a principled way for model selection without labels. We computed the held-out (10%) NLL. We found that scETM is robust to different architectures in terms of the NLL (Supplementary Table 2). We also found that ARI and NLL are modestly negatively correlated on the TM dataset (Supplementary Fig. 2), implying an agreement between the two metrics although this might not be always the case since it highly depends on the cell type labels and the data quality.

To further verify the clustering performance and validate our evaluation metrics, we visualized the cell embeddings using Uniform Manifold Approximation and Projection (UMAP)[39] for some of the datasets (Supplementary Figs. 3,4,5,6,7,8 and Fig. 2). Altogether, these results support that scETM effectively captures cell-type-specific information, while accounting for artefacts arising from individual or technological variations.

**Batch overcorrection analysis**. Some methods may risk overcorrecting batch effects and fail to capture some aspect of biological variations. In the above analysis, we observe that some methods such as LIGER conferred competitive kBET but low ARI, suggesting potential overcorrection of batch effects. To experiment the extent of batch overcorrection by each method, we conducted two experiments using two datasets, namely the Human Pancreas (HP) dataset[7] and the Mouse Retina (MR) dataset[36,37].

For the HP data, we manually removed beta cells from all 5 batches except for batch CelSeq2, resulting in the cell type distributions shown in Supplementary Table 4. We expect that, if a method is guilty of batch-effect overcorrection, it would assign

**Table 2 Batch correction performance on data integration tasks.** The batch correction performance is measured by kBET[18]. More details are described in Table 1 caption and Methods.

|  | MP | HP | TM | AD | MDD | MR |
|---|---|---|---|---|---|---|
| Harmony | 0.390 | 0.342 | 0.148 | 0.518 | 0.440 | 0.063 |
| Scanorama | 0.439 | 0.155 | 0.001 | 0.286 | 0.202 | 0.063 |
| Seurat | 0.538 | 0.381 | 0.281 | 0.195 | 0.085 | 0.053 |
| scVAE-GM | 0.233 | NA | NA | 0.073 | 0.051 | 0.033 |
| scVI | 0.516 | 0.140 | 0.056 | 0.434 | 0.313 | 0.073 |
| LIGER | 0.602 | 0.660 | 0.374 | 0.771 | 0.716 | 0.176 |
| scVI-LD | 0.148 | 0.034 | 0.069 | 0.476 | 0.277 | 0.022 |
| scETM+adv | 0.546 | 0.443 | 0.144 | 0.627 | 0.570 | 0.141 |
| scETM | 0.270 | 0.163 | 0.096 | 0.278 | 0.228 | 0.066 |
| scETM $-\lambda$ | 0.217 | 0.000 | 0.000 | 0.122 | 0.066 | 0.017 |
| Batch Effect | Strain | Technology | Technology | Individual | Individual | Studies |

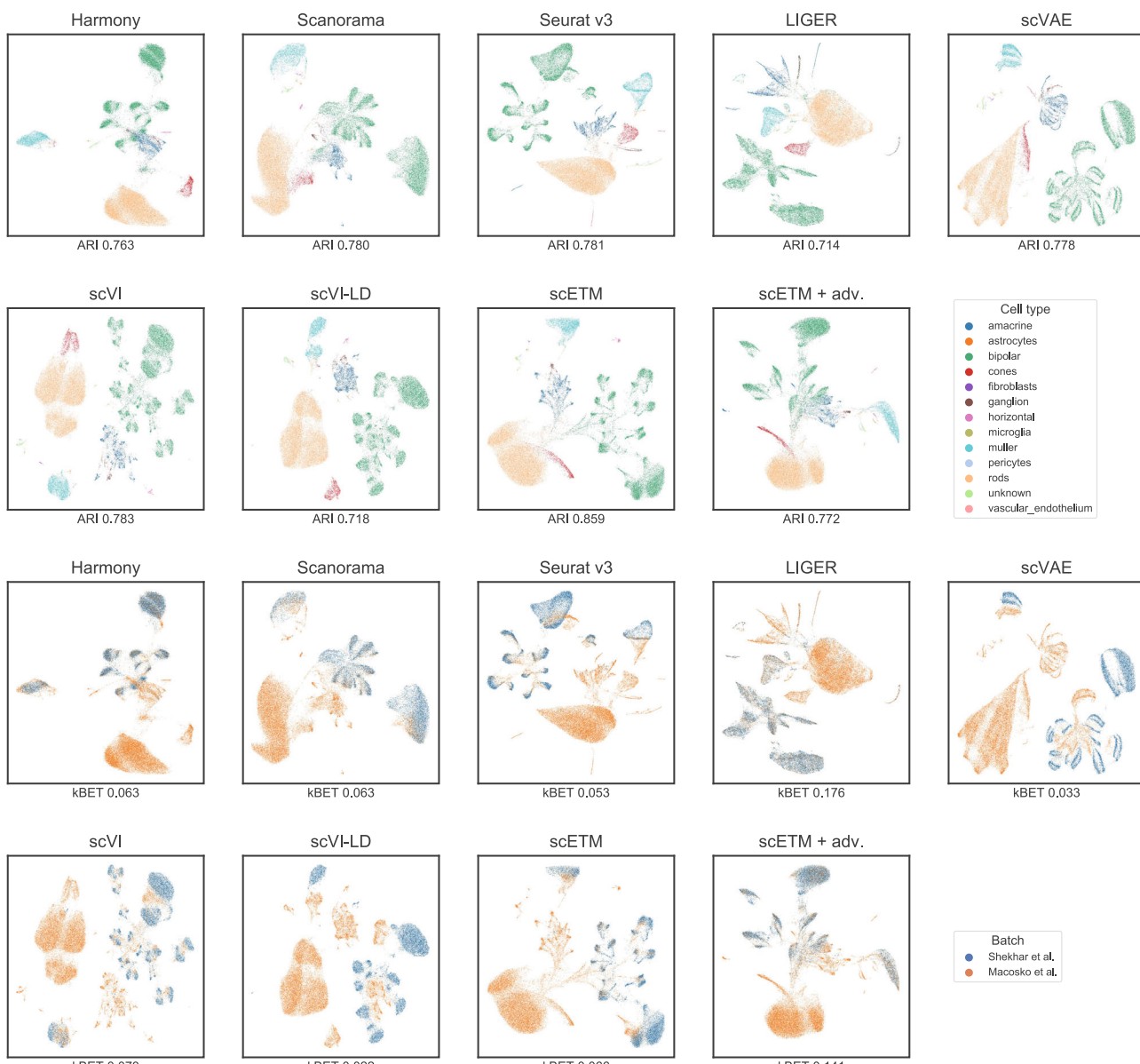

**Fig. 2 Integration and batch correction on the Mouse Retina dataset.** Each panel shows the Mouse Retina cell clusters using UMAP based on the cell embeddings obtained by each of the 9 methods. The cells are colored by cell types in the first two rows and by batches, which are the two source studies, in the last two rows. The ARI and kBET scores of each method are shown below each plot. UMAP visualization for the other 5 datasets are illustrated in Supplementary Figs. 3, 4, 5, 6, 7.

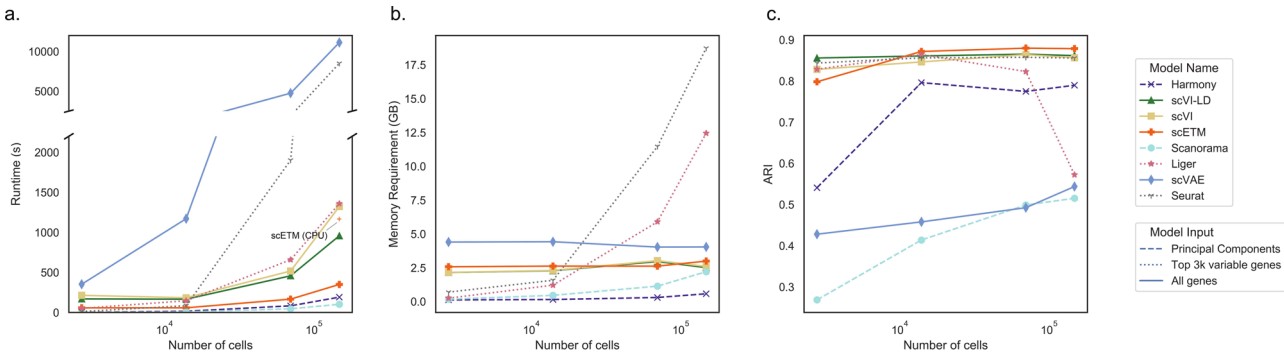

**Fig. 3 Benchmark of the efficiency and scalability of the seven scRNA-seq clustering algorithms.** The line styles in the plot indicate model inputs. The number of genes was fixed to 3000 in this experiment. We increased the number of cells randomly sampled from the combined AD and MDD dataset and evaluated the performance of each method in terms of: **a** runtime, **b** memory usage, and **c** adjusted rand index (ARI). The run-time of scETM on CPU is annotated on the left panel.

beta cells to other non-beta cell clusters in the latent space by forcing the alignment of different batches. Consequently, such methods would have poor clustering scores. We evaluated all of the methods on this dataset using 3 metrics: ARI, kBET, and average silhouette width (ASW)[40]. Briefly, higher ASW indicates larger distances between cell types and lower distance within cell type in the cell embedding space (Methods). We measured the overall ASW as well as the ASW for only the beta cells (i.e., ASW-beta). The results are summarized in Supplementary Table 5. We observed that scETM struck a good balance between discriminating cell types (ARI: 0.9298; ASW: 0.3525; ASW-beta: 0.5370) and integrating different batches (kBET: 0.1247). Adding the adversarial loss to the scETM (i.e., scETM+adv) increased kBET from 0.1247 to 0.3445 (while maintaining ARI above 0.92) but greatly compromised ASW-beta (0.0045), suggesting a more aggressive overcorrection. Similarly, LIGER performed the best in kBET (0.5978) but conferred a much lower ARI (0.8476) and low ASW-beta (0.0912), indicating a more severe overcorrection of the batch effects (i.e., mixing beta cells with other cells).

We then visualized the clustering by UMAP to examine how the beta cells are assigned to different clusters (Supplementary Fig. 9). We found that beta cells (colored in red) were clustered separately by scETM from other cell types. In contrast, beta cells were mixed up with other cell types by methods including Harmony and LIGER, which overcorrected the batch effects when integrating the five batches. Visually, we also observe that scETM +adv method moves the beta cell cluster closer to the alpha cell cluster, confirming a higher level of overcorrection compared to scETM.

The MR dataset is a collection of two independent studies on mouse retina[36,37]. Here we consider the two source studies as two batches, hereafter referred to as the Macosko batch and the Shekhar batch. Many cell types are uniquely present in the Macosko batch (Supplementary Table 6). There is also a large difference in the cell proportion between the two batches. In particular, rods is only 0.35% in Shekhar but 65% in Macosko. In this scenario, we expect that methods that overcorrect the batch effects would tend to mix rods with cells of other cell types from the Shekhar batch, resulting in low ARI and high kBET. On the contrary, a desirable integration method would strike a balance between the ARI (or ASW) and kBET on this combined dataset. Therefore, this setup imposes a great challenge on the integration methods.

Overall, scETM achieved the highest ARI (0.859), reasonable ASW (0.2873), and modest kBET (0.0656), indicating its ability to capture the true biology from the data without over-correcting the batch effects (Supplementary Table 7). In contrast, LIGER is

more aggressive in its batch correction, resulting in the highest kBET score of 0.176 but lowest ARI score of 0.714. We further investigated the extent of improving kBET while maintaining a high ARI score with scETM+adv. Indeed, scETM+adv conferred an increased kBET of 0.1410 and a reasonably high ARI score of 0.7720. Visualizing the clustering of each method using UMAP (Fig. 2) confirms the quantitative clustering results.

Incidentally, we also notice that scETM is not sensitive to the 669 doublelets or contaminants, all of which were from the Shekhar batch (Supplementary Table 8). In contrast, if we did not filter out the doublets/contaminants, the performance of LIGER and Seurat degrades drastically possibly due to batch over-correction or failing to integrate the same cell types from different batches together.

**Scalability**. A key advantage of scETM is its high scalability and efficiency. We demonstrated this by comparing the run-time, memory usage, and clustering performance of the state-of-the-art models using their recommended pipelines when integrating a merged dataset consisting of cells from the MDD and AD data-sets (Methods). Because of the simple model design and efficient implementation (e.g., sparse matrix representation, multi-threaded data retrieval, etc; Discussion), scETM achieved the shortest run-time among all deep-learning based models (Fig. 3a). Specifically, on the largest dataset (148,247 cells), scETM ran 3–4 times faster than scVI and scVI-LD, and over 10 times faster than scVAE-GM. We note that the run-time largely depends on the implementation rather than the network architectures and loss functions in these deep-learning methods. Harmony and Sca-norama were the only methods faster than scETM, yet they both operate on a hundred principal components at most. Although for comparison purpose we used the top 3000 most variable genes for all of the methods, scETM can easily scale to all of the genes, which is more desirable because the resulting model can gen-eralize to other datasets.

Because of the amortized stochastic variational inference[32,41,42], scETM in principle takes linear run-time and constant memory with respect to the sample size per training epoch. The use of multi-threaded data loader to streamline the random minibatch retrieval and loading further speed-up the training process in practice. In contrast, the memory requirement of Seurat increases rapidly with the number of cells, due to the vast numbers of plausible anchor cell pairs in the two brain datasets (Fig. 3b). In terms of clustering accuracy, scETM consistently confers compe-titive performance, whereas Harmony and Scanorama perform unstably as dataset sizes vary (Fig. 3c). UMAP visual inspection of

scVAE embeddings suggests that scVAE likely suffers from under-correction of batch effects (Supplementary Fig. 8). The sudden drop of LIGER's clustering performance in the largest benchmark dataset may be due to batch overcorrection.

Although it has been widely accepted by the deep-learning community that computing using Graphical Processing Units (GPUs) results in ~ 10 × speed-up over computing using CPU, the adoption of GPUs in the computational biology community is beginning to catch up. In our non-exhaustive experiment on the Mouse Pancreas dataset, training scETM for 1000 steps on the 6-core Core i7 10750H CPU requires 650 s (Fig. 3a), while with an Nvidia RTX 2070 laptop GPU it only takes 50 s—a 13 × speed-up over the CPU computer.

**Transfer learning across single-cell datasets.** A prominent feature of scETM is that its parameters, hence the knowledge of modeling scRNA-seq data, are transferable across datasets. Specifically, as part of the scETM, the encoder trained on a reference scRNA-seq dataset can be applied to infer cell topic mixture of a target scRNA-seq dataset (Fig. 1b), regardless of whether the two datasets share the same cell types. As an example, we trained an scETM model on the Tabula Muris FACS dataset (TM (FACS)) (which is a subset of the TM dataset) from a multi-organ mouse single-cell atlas, and evaluated it using the MP data, which only contains mouse pancreatic islet cells. Although the two datasets were obtained using different sequencing technologies in two independent studies, the model yielded an encouragingly high ARI score of 0.941, considering that a model directly trained on MP achieves ARI 0.946. Interestingly, in the UMAP plot, the TM (FACS)-pretrained model placed B cells, T cells and macrophages far away from other clusters and separated B cells and T cells from macrophages, which is not observed in the model directly trained on MP (Supplementary Figs. 3,4,5,6,7; Supplementary Table 9). We repeated the same experiment three times with different random seeds and observed consistently that B and T cells are close to each other and distant from macrophages (Supplementary Fig. 10). We also experimented transfer learning by first training scETM on TM (FACS) with pancreas removed and then applied to MP dataset. The performance decreased but is still reasonably good (Supplementary Table 9), demonstrating scETM's ability to transfer knowledge across tissues.

Encouraged by the above results, we then performed a comprehensive set of cross-tissue and cross-species transfer-learning analysis with 6 tasks (Methods): (1) Transfer between the TM (FACS) and the MP dataset (including MP → TM (FACS)); (2) Transfer between the Human Pancreas (HP) dataset and the Mouse Pancreas (MP) dataset; (3) Transfer between the Human primary motor cortex (M1C) (HumM1C) dataset and the Mouse primary motor area (MusMOp) dataset both obtained from the Allen Brain Map data portal[43]. We chose to transfer between the human M1C and mouse MOp because of the high number of shared cell types between the brain regions of the two species. The batches for HumM1C are the two post-mortem human brain M1 specimens and the two mice for MusMOp. Note that in these transfer-learning tasks ($A \rightarrow B$) we only corrected batch effects during the training on the source data $A$ but not during the transfer to the target data $B$.

As a comparison, we evaluated and visualized the clustering results in all 6 transfer-learning tasks using scETM, scVI-LD, and scVI (Fig. 4; Supplementary Tables 10 and 11). Overall, scETM achieved the highest ARI across all tasks and competitive kBET scores. In particular, scETM trained on TM (FACS) on heterogeneous tissues clustered much better the MP cells (ARI: 0.941; kBET: 0.339) than scVI (ARI: 0.484; kBET: 0.257) and scVI-LD (ARI: 0.398; kBET: 0.256). Remarkably, scETM trained

only on the MP dataset can cluster reasonably well the much larger TM single-cell data, which were collected from diverse primary tissues including pancreas. This implies that scETM does not merely learn cell-type-specific signatures but also the underlying transcriptional programs that are generalizable to unseen tissues.

In cross-species transfer learning between HP and MP, scETM captured better the conserved pancreas functions compared to scVI and scVI-LD (Fig. 4; Supplementary Table 10). On the other hand, cross-species transfer between MusMOp and HumM1C is a much more challenging task due to the evolutionarily divergent functions of the brains between the two species. Nonetheless, scETM conferred a much higher ARI of 0.696 for the MusMOp → HumM1C transfer and ARI of 0.167 for the HumM1C → MusMOp transfer. In contrast, scVI-LD and scVI did not work well on these tasks with ARI scores lower than 0.1. Since they cannot separate cells by cell types, all cells are mixed together, leading to a high kBET score. The improvements achieved by scETM over scVI(-LD) are possibly attributable to the simpler linear batch correction on the source data, jointly learning topic and gene embedding, and the topic-modeling formalism, which together lead to an encoder network that is better at capturing the transferable cellular programs.

**Pathway enrichment analysis of scETM topics.** We next investigated whether the scETM-inferred topics are biologically relevant in terms of known gene pathways in human. One approach would be to arbitrarily choose a number of top genes under each topic and test for pathway enrichment using hypergeometric tests. This approach works well when there are asymptotic $p$-values at the individual gene level. In our case, each gene is characterized by the topic scores, making it difficult to systematically choose the number of top genes per topic. To this end, we resorted to Gene Set Enrichment Analysis (GSEA)[44]. Briefly, we calculated the maximum running sum of the enrichment scores with respect to a query gene set by going down the gene list that is sorted in the decreasing order by a given topic distribution $\beta_k$ (Methods). For each dataset, we trained a scETM with 100 topics.

For the HP dataset, each topic detected many significantly enriched pathways with Benjamini–Hochberg False Discover Rate (FDR) < 0.01 (Fig. 5a). Many of them are relevant to pancreas functions, including insulin processing (Fig. 5b), insulin receptor recycling, insulin glucose pathway, pancreatic cancer, etc (Supplementary Table 12). As scETM jointly learns both the gene embeddings and topic embeddings, we can visualize both the genes and topics in the same embedding space via UMAP (Fig. 5c). Indeed, we observe a strong co-localization of the genes in Insulin Processing pathway and the corresponding enriched topic (i.e., Topic 54).

For the AD dataset, we found topics enriched for Reactome Amyloid Fiber Formation, KEGG AD, and Deregulated CDK5 triggers multiple neurodegenerative in AD (Supplementary Fig. 11 and Supplementary Table 13). For MDD dataset, we found enrichment for substance/drug induced depressive disorder (Supplementary Figs. 12, 13 and Supplementary Table 14). The full GSEA enrichment results for all 3 datasets are listed in Supplementary Data 1. As a comparison, we also performed GSEA over the 100 gene loadings learned by scVI-LD (matching the 100 topics in the scETM) on these 3 datasets but found fewer relevant distinct gene sets (Supplementary Tables 12, 13 and Supplementary Table 14) or weaker statistical enrichments by GSEA (Supplementary Fig. 14).

**Differential scETM topics in disease conditions and cell types.** We sought to discover scETM topics that are condition-specific

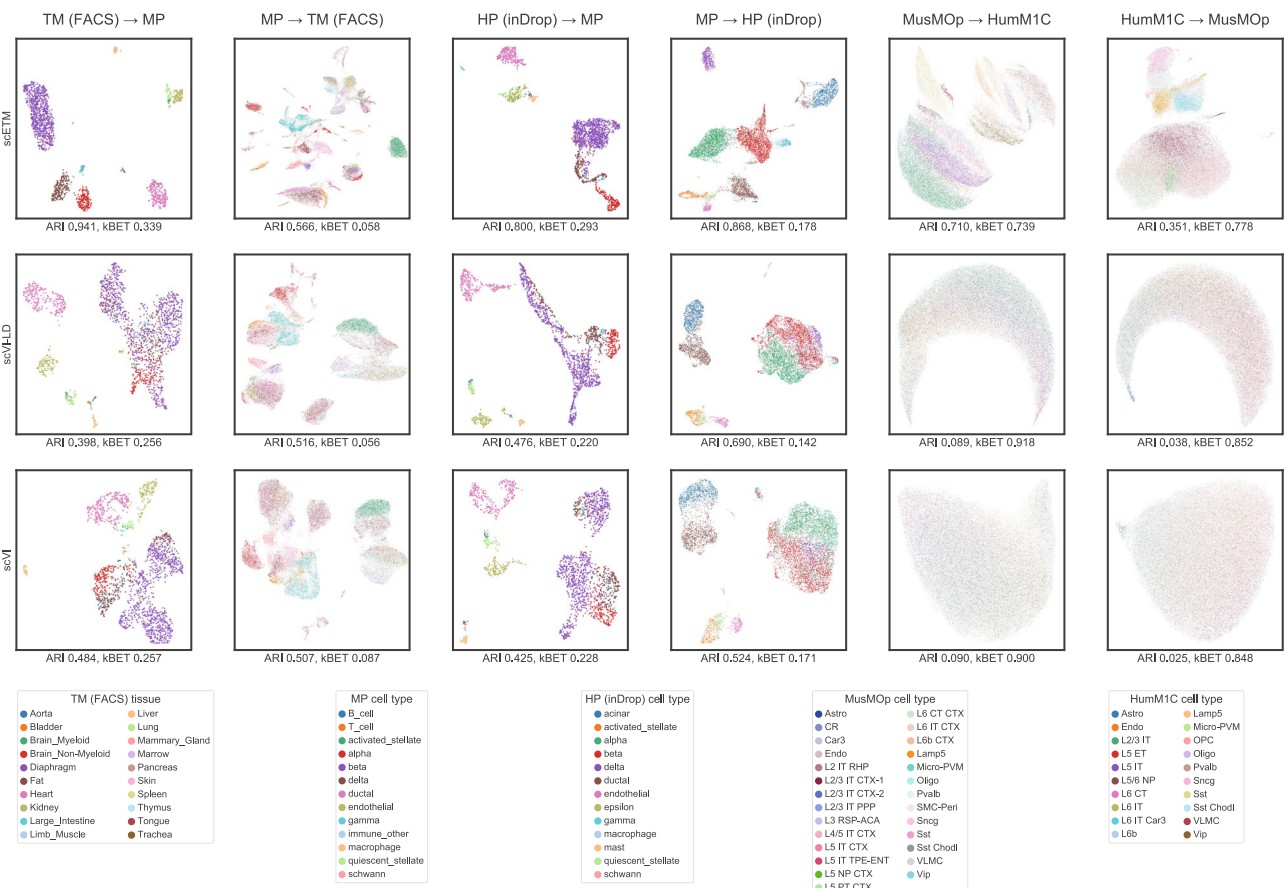

**Fig. 4 Cross-tissue and cross-species zero-shot transfer learning.** Each panel displays the UMAP visualization of cells by training on dataset A and the applied to dataset B (i.e., A → B). In total, we performed 6 transfer-learning tasks: TM (FACS) ↔ MP, HP (inDrop) ↔ MP, MusMOp ↔ HumM1C. For each task, we evaluated scETM, scVI-LD, or scVI, which are the rows in the above figure. The cells are colored by tissues or cell types as indicated in the legend. The corresponding ARI and kBET are indicated below each panel. For MP → TM (FACS), we evaluated the ARI based on the 92 cell types although we colored the cells by the 20 tissues of origin instead of their cell types because of the large number of cell types. Abbreviations: TM (FACS): tabula muris sequenced with fluorescence-activated single-cell sorting; MP: mouse pancreas; HP (inDrop): human pancreas sequenced with InDrop technology; HumM1C: human primary motor cortex (from Allen Brain map); MusMOp: mouse primary motor area (from Allen Brain map).

or cell-type specific. Starting with the AD dataset, we found that the scETM-learned topics are highly selective of cell-type marker genes (Fig. 6a) and highly discriminative of cell types (Fig. 6b). To detect disease signatures, we separated the cells into the ones derived from the 24 AD subjects and the ones from the 24 control subjects. We then performed permutation tests to evaluate whether the two cell groups exhibit significant differences in terms of their topic expression (Methods). Topic 12 and 58 are differentially expressed in the AD cells and control cells (Fig. 6c, d; permutation test $p$-value = 1e-5). Interestingly, topic 58 is also highly enriched for mitochondrial genes. Indeed, it is known that $\beta$-amyloids selectively build up in the mitochondria in the cells of AD-affected brains[45]. For the MDD dataset, topics 1, 52, 68, 70, 86 exhibit differential expressions between the suicidal group and the healthy group (Supplementary Fig. 18c) and interesting neurological pathway enrichments (Supplementary Table 14).

We also identified several cell-type-specific scETM topics from the HP, AD, and MDD datasets. In HP, topics or metagenes 20, 45, 99 are upregulated in acinar cells, topic 12 upregulated in macrophage, topic 52 upregulated in delta, and topics 30 and 37 are upregulated in more than one cell types, including endothelial, stellate and others (Supplementary Fig. 15). In AD, as shown by both the cell topic mixture heatmap and the differential expression analysis (Fig. 6b), topics 19, 35, 50, 69, 97 are upregulated in oligodendrocytes, micro/macroglia, astrocytes,

endothelial cells, and oligodendrocyte progenitor cells (OPCs) respectively (permutation test $p$-value = 1e-5; Fig. 6b, Supplementary Fig. 16). Interestingly, two subpopulations of cells from the oligodendrocytes (Oli) and excitatory (Ex) exhibit high expression of topics 12 and 58, respectively, and are primarily AD cells (Supplementary Fig. 17). Among them, there is also a strong enrichment for the female subjects, which is consistent with the original finding[35].

For MDD, topics 1, 20, 59, 64, and 72 are upregulated in astrocytes, oligodendrocytes, micro/macroglia, endothelial cells, and OPCs, respectively (Supplementary Fig. 18c). This is consistent with the heatmap pattern (Supplementary Fig. 18b). Several topics are dominated by long non-coding RNAs (lincRNAs) (Supplementary Fig. 18a). While previous studies have suggested that lincRNAs can be cell-type-specific[46], it remains difficult to interpret them[47]. We further experimented the enrichment using only the protein coding genes, but did not find significantly more marker genes among the top 10 genes per topic (Supplementary Fig. 19).

**Pathway-informed scETM topics**. To further improve topic interpretability, we incorporated the known pathway information to guide the learning of the topic embeddings (Fig. 7a). We denoted this scETM variant as the pathway-informed scETM or

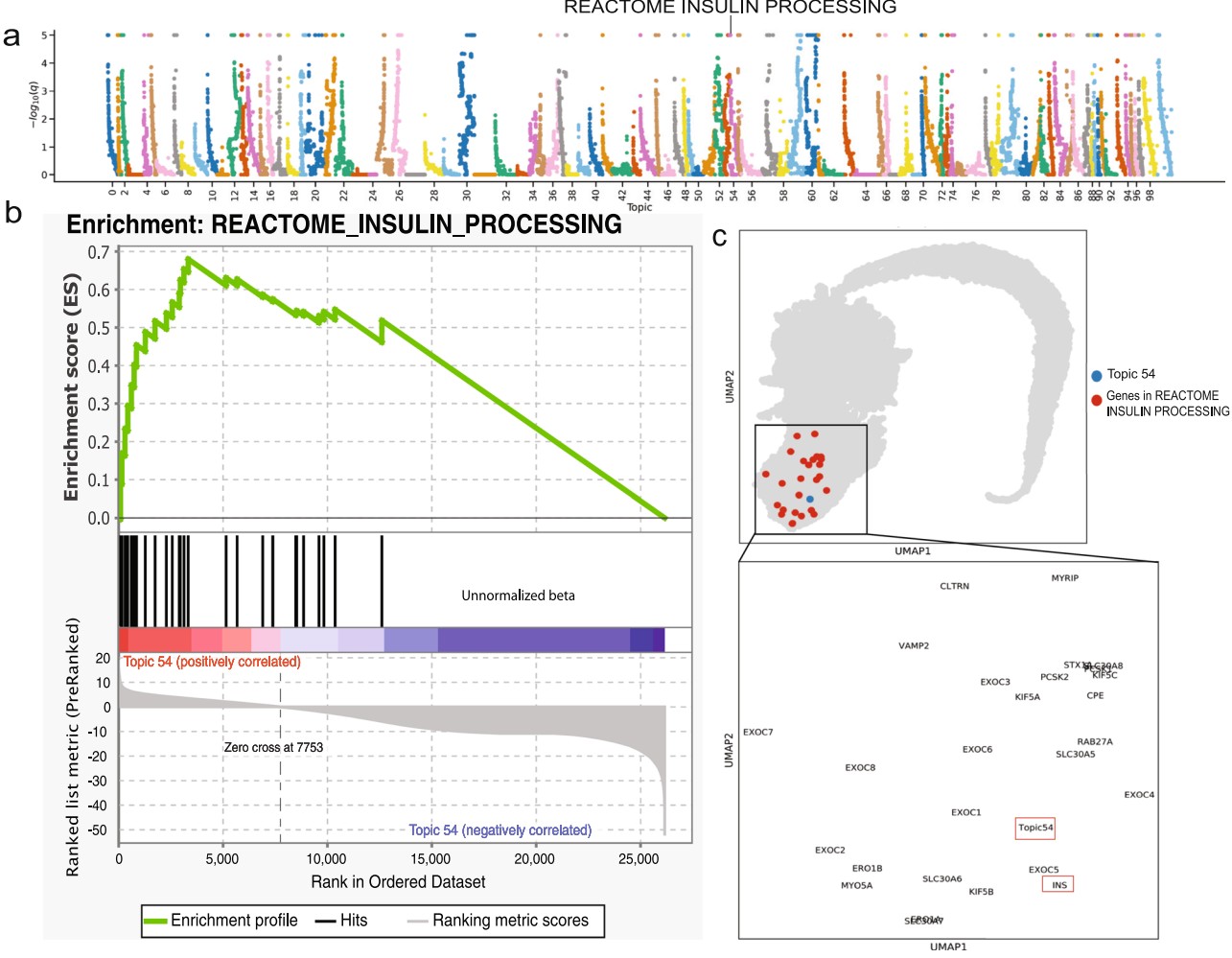

**Fig. 5 Gene set enrichment analysis of the Human Pancreas dataset. a** Manhattan plot of the GSEA results on the 100 scETM topics learned from the HP dataset. The x-axis is the topic and the y-axis is the -log q-value from the permutation test corrected for multiple testings by the BH-method. The dots correspond to the tested GSEA gene sets. The maximum -log q-value is capped at 5. **b** Leading edge analysis of Insulin Processing (IP) pathway using Topic 54. The genes are ranked by the topic score (i.e., the unnormalized topic mixture beta) under Topic 54. The running sum enrichment score was calculated by GSEA. The black bars in the middle indicate the genes that are found in the IP pathway. Topic 54 is significantly enriched in Insulin Processing pathway (GSEA permutation test BH-adjusted q-value = 0). **c** UMAP visualization of the gene and topic embeddings learned from the HP dataset. Genes in IP are colored in red and Topic 54 in blue. The inset box displays a magnified view of the cluster zooming into the IP pathway genes (including *Insulin* (*INS*)) that are near Topic 54 in the embedding space.

p-scETM. In particular, we fixed the gene embedding $\rho$ to a pathways-by-genes matrix obtained from pathDIP4 database[48]. We then learned only the topics embedding $\alpha$, which provides direct associations between topics and pathways (Methods). We tested p-scETM on the HP, AD and MDD datasets. Without compromising the clustering performance (Supplementary Table 15), p-scETM learned functionally meaningful topic embeddings (Supplementary Fig. 20; Supplementary Tables 16 and 17). In the HP topic embeddings, we found Insulin Signaling, Nutrient Digestion and Metabolism to be the top pathways among several topics (Supplementary Fig. 20a). In the MDD topic embeddings, the top pathway associated with Topic 40, Beta-2 Adrenergic Receptor Signaling, was also enriched in a MDD genome-wide association studies[49]. In the AD topic embeddings, we found the association between Topic 9 and Alzheimer Disease-Amyloid Secretase pathway.

To further demonstrate the utility of p-scETM, we also used 7481 gene ontology biological process (GO-BP) terms[50,51] as the

fixed gene embedding, which learns the topics-by-GOs topic embedding from each dataset. Under each topic, we selected the top 5 high-scoring GO-BP terms to examine their relevance to the target tissue or disease (Fig. 7b and Supplementary Data 2). For the HP dataset, Negative Regulation of Type B Pancreatic Cell (GO:2000675) and Regulation of Pancreatic Juice Secretion (GO:0090186) are among the top GO-BP terms for Topics 27 and 68, respectively. For the AD dataset, Amyloid Precursor Protein Biosynthetic Process (GO:0042983) is among the top 5 GO-BP terms under Topic 40. For the MDD dataset, similar top GO-BP terms were found among topics learned using all of the genes and using only the coding gene. Many topics exhibit high embedding scores for neuronal functions including Neuronal Signal Transduction (GO:0023041), Central Nervous System Projection Neuron Axonogenesis (GO:0021952), and Branchio-motor Neuron Axon Guidance (GO:0021785). Interestingly, Topic 98 in MDD—coding genes only and Topics 22, 51 in MDD—all genes involve Adenylate Cyclase Modulating

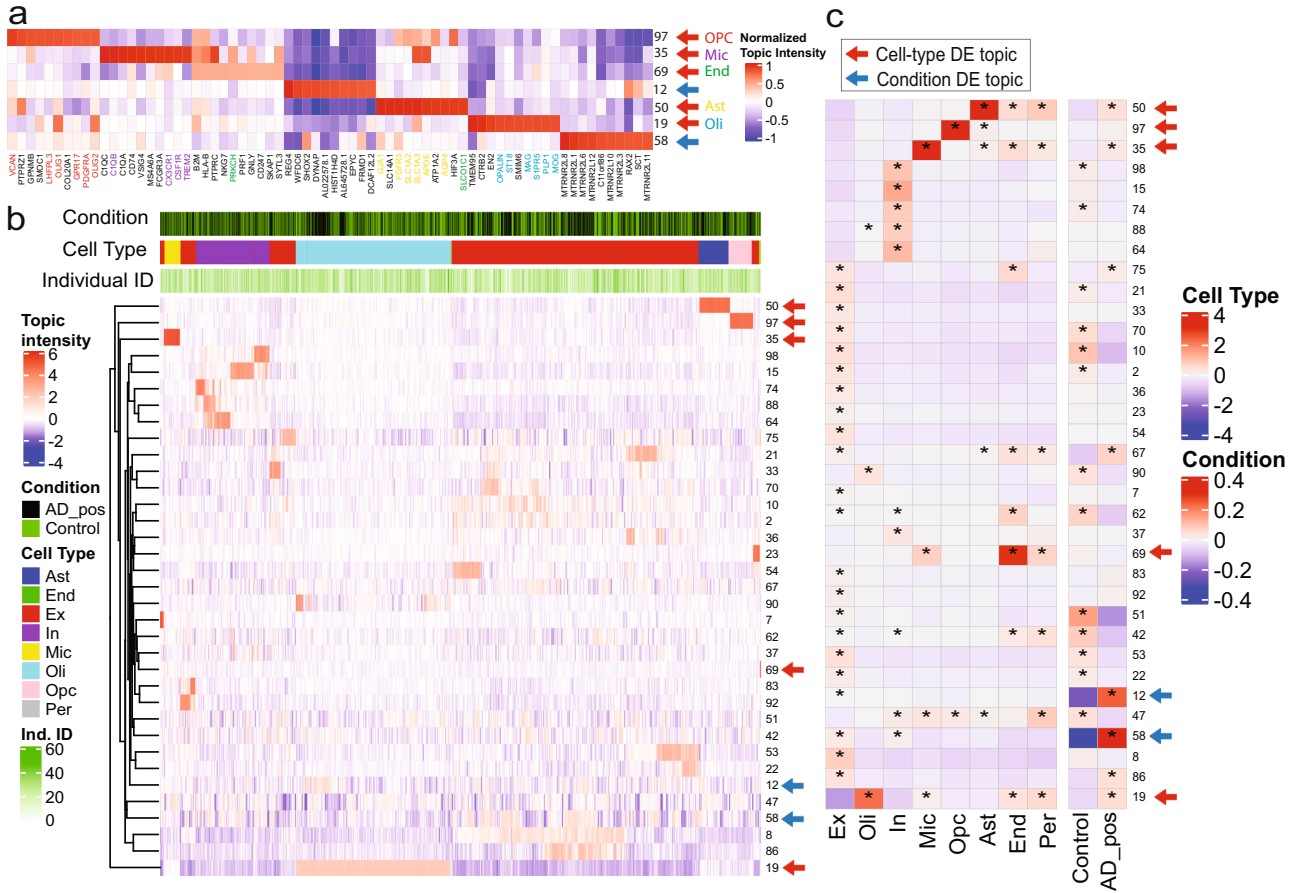

**Fig. 6 scETM topic embeddings of the Alzheimer's Disease snRNA-seq dataset. a** Gene-topic heatmap. The top genes that are known as cell-type marker genes based on PanglaoDB are highlighted. For visualization purposes, we divided the topic values by the maximum absolute value within the same topic. Only the differential topics with respect to cell-type or AD were shown. **b** Topics intensity of cells ($n = 10,000$) sub-sampled from the AD dataset. Topic intensities shown here are the Gaussian mean before applying softmax. Only the select topics with the sum of absolute values >1500 across all sampled cells are shown. The three color bars show disease conditions, cell types, and batch identifiers (i.e., subject IDs). **c** Differential expression analysis of topics across the eight cell types and two clinical conditions. Colors indicate mean differences between cell groups with and without a certain label (cell-type or condition). Asterisks indicate Bonferroni-corrected empirical $p$-value < 0.05 for 100,000 one-sided permutation tests of upregulated topics in each cell-type and disease labels.

G-protein Coupled Receptors (GPCRs) Signaling (GO:0007188), which is the target of several recently-developed antidepressant drugs[52].

## Discussion

As scRNA-seq technologies become increasingly affordable and accessible, large-scale datasets have emerged. This challenges traditional statistical approaches and calls for transferable, scalable, and interpretable representation learning methods to mine the latent biological knowledge from the vast amount of scRNA-seq data. To address these challenges, we developed scETM and demonstrated its state-of-the-art performance on several unsupervised learning tasks across diverse scRNA-seq datasets. scETM demonstrates excellent capabilities of batch-effect correction and knowledge transfer across datasets.

In terms of integrating multiple scRNA-seq data from different technologies, experimental batches, or studies, we introduce a simple batch-effect bias term to correct for non-biological effects. This in general improves the cell clustering and topic quality. When using the original ETM[33], we observed that ubiquitously expressed genes such as *MALAT1* tended to appear among the top genes in several topics. Our scETM corrects the background gene expressions by the gene-dependent and batch-dependent

intercepts. As a result, the ubiquitously expressed genes do not dominate all topics from scETM. We also introduced a more aggressive batch correction strategy by adversarial network loss, which shows improved kBET with small trade-off for the ARI in most datasets.

In terms of scalability, although scETM is similar to other existing VAE models in terms of theoretical time and space complexity, we emphasize that implementation is also very important, especially for deep-learning models. For example, scVAE-GM[11] is much slower and more memory consuming than scVI[6], while they are very similar VAE models. One of the main speedups provided by scETM comes from our implementation of a multi-threaded data loader for minibatches of cells, which does not need to be re-initialized at every training epoch as the standard PyTorch DataLoader. Compared to scVI and scVI-LD, the normalized counts in both the encoder input and the reconstruction loss used by scETM remove the need to infer the cell-specific library size variable, and the simpler categorical likelihood choice also helps reduce the computational time.

In terms of transferrability, many existing integration methods require running on both reference and query datasets to perform post hoc analyses such as joint clustering and label transfer[7,9,10,28]. In contrast, our method enables a direct or zero-shot knowledge transfer of the pretrained encoder network

## a. Pathway-informed scETM (p-scETM)

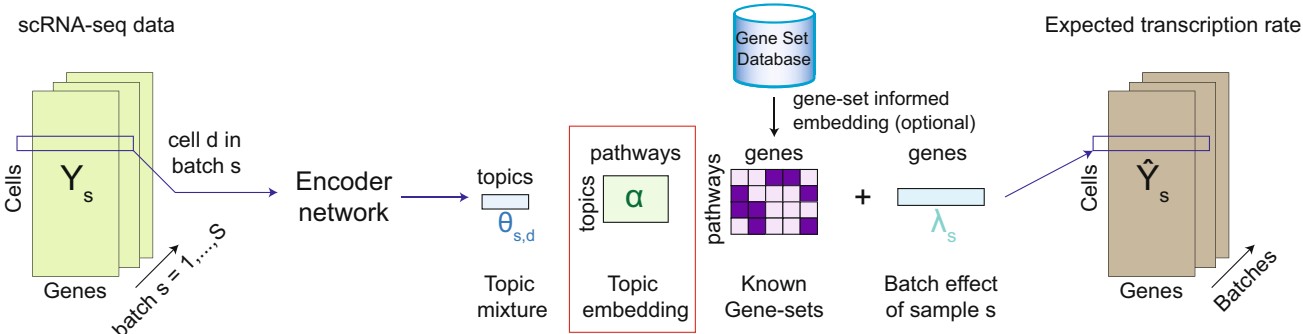

## b. topic embeddings from the four scRNA-seq datasets

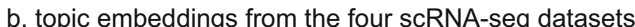

**Fig. 7 Pathway-topics embeddings learned by the pathway-informed scETM (p-scETM). a** p-scETM overview. Pathways information as pathways-by-genes are provided as the gene embedding in the linear decoder. The learned topic embedding is the direct association between the topics and pathways. **b** The pathway-topics heatmap of top 5 pathways in selected topics. Here, the pathways are the Gene Ontology-Biological Processes (GO-BP) terms. For the HP dataset, GO-BP terms whose names include the keywords insulin or pancreatic were highlighted. For the AD dataset, GO-BP terms whose names include the keywords amyloid or alzheimer were highlighted. For MDD—all genes and MDD—coding genes only, GO-BP terms whose names include the keywords neuron or G-protein were highlighted.

parameters learned from a reference dataset in annotating a new target dataset without further training. We demonstrated this important aspect in cross-technology, cross-tissue, and cross-species applications, for which we achieved superior performance compared to the state-of-the-art methods.

In terms of interpretability, our quantitative experiments showed that scETM identified more relevant pathways than scVI-LD. Qualitative experiments also show that scETM topics preserve cell functional and cell-type-specific biological signals implicated in the single-cell transcriptomes. By seamlessly incorporating the known pathway information in the gene embedding, p-scETM finds biologically and pathologically important pathways by directly learning the association between the topics with the pathways via the topic embedding. Recently proposed by[53], single-cell Hierarchical Poisson Factor (scHPF) model applies hierarchical Poisson factorization to discover interpretable gene expression signatures in an attempt to address the interpretability challenge. However, compared to our model, scHPF lacks the flexibility in learning the gene embedding and incorporating existing pathway knowledge, and is not designed to account for batch effects. Moreover, scETM has the benefits of both flexibility in the neural-network encoder and the interpretability in the linear decoder.

As future work, we will extend scETM in several directions. To further improve batch correction, as our current model only considers a single categorical batch variable, we can extend it to correct for multiple categorical batch variables. For a small number of categorical batch variables, we may use several sets of batch intercept terms to model them. For hierarchical batch variables, we may use a tree of batch intercept terms. For numerical batch effects such as subject age, one way is to convert them into categorical batch variables by numerical ranges. When the number of batch variables become larger, we consider three strategies. First, we can add the batch variables as the covariates in the linear regression on the gene expression and fit the linear coefficients each corresponding to a sample-dependent batch variable. Second, we can factorize the batches-by-genes into batches-by-factors and factors-by-genes. Learning the two matrices will be similar to the scETM algorithm. Third, we can extend our current scETM+adv to correct for both categorical and continuous batch variables with a discriminator network, which predicts batch effects using the encoder-generated cell topic mixture[38].

To further improve data integration, we can extend scETM to a multi-omic integration method, which can integrate scRNA-seq plus other omics such as protein expression measured in the same cells as scRNA-seq[54] or scATAC-seq measured in different cells but the same biological system[7]. In these applications, multi-modality over different omics will need to be considered to capture the intrinsic technical and biological variance of each omic while borrowing information among them.

To further improve interpretability, the original ETM used pretrained word embedding from word2vec[55] on a larger reference corpus such as Wikipedia to improve topic quality on modeling the target documents[33]. Similarly, although we demonstrated the use of existing pathway information in p-scETM, we can also pretrain our gene embeddings on PubMed articles, gene regulatory network, protein-protein interactions, or Gene Ontology graph using either gene2vec[56] or more general graph embedding approaches[57,58]. We expect that the gene embedding pretrained from these (structured) knowledge graphs will further improve the efficiency and interpretability of scETM.

Together, scETM serves as a unified and highly scalable framework for integrative analysis of large-scale single-cell transcriptomes across multiple datasets. Compared to existing methods, scETM offers consistently competitive performance in

data integration, transfer learning, scalability, and interpretability. The simple Bayesian model design in scETM also provides a highly expandable framework for future developments.

## Methods

**scETM data generative process.** To model scRNA-seq data distribution, we take a topic-modeling approach[59]. In our framework, each cell is considered as a document, each scRNA-seq read (or UMI) as a token in the document, and the gene that gives rise to the read (or UMI) is considered as a word from the vocabulary of size $V$. We assume that each cell can be represented as a mixture of latent cell types, which are commonly referred to as the latent topics. The original LDA model[59] defines a fixed set of $K$ independent Dirichlet distributions $\boldsymbol{\beta}$ over a vocabulary of size $V$. Following the ETM model[33], here we decompose the unnormalized topic distribution $\boldsymbol{\beta}^* \in \mathbb{R}^{K \times V}$ into the topic embedding $\boldsymbol{\alpha} \in \mathbb{R}^{K \times L}$ and gene embedding $\boldsymbol{\rho} \in \mathbb{R}^{L \times V}$, where $L$ denotes the size of the embedding space. Therefore, the unnormalized probability of a gene belonging to a topic is proportional to the dot product between the topic embedding matrix and the gene embedding matrix. Formally, the data generating process of each scRNA-seq profile $d$ is:

1. Draw a latent cell type proportion $\boldsymbol{\theta}_d$ for a cell $d$ from logistic normal $\boldsymbol{\theta}_d \sim \mathcal{LN}(0, \mathbf{I})$:

$$\boldsymbol{\delta}_d \sim \mathcal{N}(0, \mathbf{I}), \quad \boldsymbol{\theta}_d = \text{softmax}(\boldsymbol{\delta}_d) = \frac{\exp(\delta_{d,k})}{\sum_{k=1}^{K} \exp(\delta_{d,k})} \quad (1)$$

2. For each gene read (or UMI) $w_{i,d}$ in cell $d$, draw gene $g$ from a categorical distribution $\text{Cat}(\mathbf{r}_{d,\cdot})$:

$$w_{i,d} \sim \prod_g r_{d,g}^{[w_{i,d}=g]}, \quad y_{d,g} = \sum_{i=1}^{N_d} [w_{i,d} = g] \quad (2)$$

Here $N_d$ is the library size of cell $d$, $w_{i,d}$ is the index of the gene that gives rise to the $i^{th}$ read (or UMI) in cell $d$ (i.e., $[w_{i,d}=g]$), and $y_{d,g}$ is the total counts of gene $g$ in cell $d$. The transcription rate $r_{d,g}$ is parameterized as follows:

$$r_{d,g} = \frac{\exp(\hat{r}_{d,g})}{\sum_{g'} \exp(\hat{r}_{d,g'})}, \quad \hat{r}_{d,g} = \boldsymbol{\theta}_d \boldsymbol{\alpha} \boldsymbol{\rho}_g + \lambda_{s(d),g} \quad (3)$$

Here $\boldsymbol{\theta}_d$ is the $1 \times K$ cell topic mixture for cell $d$, $\boldsymbol{\alpha}$ is the global $K \times L$ cell topic embedding, $\boldsymbol{\rho}_g$ is a $L \times 1$ gene-specific embedding, and $\lambda_{s(d),g}$ is the batch-dependent and gene-specific scalar effect, where $s(d)$ indicates the batch index for cell $d$. Notably, to model the sparsity of gene expression in each cell (i.e., only a small fraction of the genes have non-zero expression), we use the softmax function to normalize the transcription rate over all of the genes.

**scETM model inference.** In scETM, we treat the latent cell topic mixture $\boldsymbol{\delta}_d$ for each cell $d$ as the only latent variable. We treat the topic embedding $\boldsymbol{\alpha}$, the gene-specific transcriptomic embedding $\boldsymbol{\rho}$, and the batch-effect $\boldsymbol{\lambda}$ as point estimates. Let $\mathbf{Y}$ be the $D \times V$ gene expression matrix for $D$ cells and $V$ genes. The posterior distribution of the latent variables $p(\boldsymbol{\delta}|\mathbf{Y})$ is intractable. Hence, we took a variational inference approach using a proposed distribution $q(\boldsymbol{\delta}_d)$ to approximate the true posterior. Specifically, we define the following proposed distribution: $q(\boldsymbol{\delta}|\mathbf{y}) = \prod_d q(\boldsymbol{\delta}_d|\mathbf{y}_d)$, where $q(\boldsymbol{\delta}_d|\mathbf{y}_d) = \boldsymbol{\mu}_d + \text{diag}(\boldsymbol{\sigma}_d) \mathcal{N}(0, I)$ and $[\boldsymbol{\mu}_d, \log \boldsymbol{\sigma}_d^2] = \text{NNET}(\bar{\mathbf{y}}_d; \mathbf{W}_\theta)$. Here $\bar{\mathbf{y}}_d$ is the normalized counts for each gene as the raw count of the gene divided by the total counts in cell $d$. The function $\text{NNET}(\mathbf{v}; \mathbf{W})$ is a two-layer feed-forward neural-network used to estimate the sufficient statistics of the proposed distribution for the cell topic mixture $\boldsymbol{\delta}_d$.

To learn the above variational parameters $\mathbf{W}_\theta$, we optimize the evidence lower bound (ELBO) of the log-likelihood, which is equivalent to minimizing the Kullback-Leibler (KL) divergence between the true posterior and the proposed distribution: $\text{ELBO} = \mathbb{E}_q[\log p(\mathbf{Y}|\Theta)] - \text{KL}[q(\Theta|\mathbf{Y})||p(\Theta)]$. The Bayesian learning is carried out by maximizing the reconstruction likelihood with regularization in the form of KL divergence of the proposed distribution $(q(\boldsymbol{\delta}_d|\mathbf{y}_d) = \mathcal{N}(\boldsymbol{\mu}_d, \text{diag}(\boldsymbol{\sigma}_d)))$ from the prior $(p(\boldsymbol{\delta}_d) = \mathcal{N}(0, \mathbf{I}))$. For computational efficiency, we optimize ELBO with respect to the variational parameters by amortized variational inference[32,41,42]. Specifically, we draw a sample of the latent variables from $q(\boldsymbol{\delta}|\mathbf{y})$ for a minibatch of cells from reparameterized Gaussian proposed distribution $q(\boldsymbol{\delta}|\mathbf{y})$[32], which has the mean and variance determined by the NNET functions. We then use those draws as the noisy estimates of the variational expectation for the ELBO. The optimization is then carried out by back-propagating the gradients into the encoder weights and the topic and gene embeddings.

**Details of training scETM.** We chose the encoder for inferring the cell topic mixture to be a 2-layer neural-network, with hidden sizes of 128, ReLU activations[60], 1D batch normalization[61], and 0.1 drop-out rate between layers. We set the gene embedding dimension to 400, and the number of topics to 50. We optimize our model with Adam Optimizer and a 0.005 learning rate. To prevent over-regularization, we start with zero weight penalty on the KL divergence and

linearly increase the weight of the KL divergence in the ELBO loss to $10^{-7}$ during the first $\frac{1}{3}$ epochs. With a minibatch size of 2000, scETM typically needs 5k-20k training steps to converge. We show that our model is robust to changes in the above hyperparameters (Supplementary Table 2). During the evaluation, we used the variational mean of the unnormalized topic mixture $\mu_d$ in $q(\delta_d|\mathbf{y}_d) = \mathcal{N}(\mu_d, \mathrm{diag}(\sigma_d))$ as the scETM cell topic mixture for cell $d$.

**scETM+adv: adversarial loss for further batch correction**. In the scETM+adv variant, we added a discriminator network (a two-layer fully-connected network) to predict batch labels using the unnormalized cell topic mixture embedding $\delta$ generated by the encoder network. This discriminator helps batch correction in an adversarial fashion. Specifically, in each training iteration, we first update the scETM parameters once by maximizing the ELBO plus the batch prediction cross-entropy loss from the discriminator, with a hyperparameter controlling the weight of the latter term. It should be noted that by maximizing the prediction loss the encoder network learns to produce batch agnostic cell embeddings. Then we update the discriminator network eight times by trying to minimize the cross-entropy loss in predicting the batch labels.

**scETM software**. We implemented scETM using the PyTorch library[62]. Our initial implementation was based on the ETM code from GitHub (adjidieng/ETM) by[33]. Since then, we completely revamped the code to substantially improve the scalability and to integrate it into the Python ecosystem. In particular, we packaged and released our code on PyPI so one can easily install the package by entering `pip install scETM` in the terminal. The package is integrated with `scanpy`[16] and `tensorboard`[63]. Users can view the cell, gene and topic embeddings interactively via tensorboard. For example, one can easily train a scETM as follows:

```
from scETM import scETM, UnsupervisedTrainer
model = scETM(adata.n_vars, adata.obs.batch_indices.nunique())
trainer = UnsupervisedTrainer(model, adata)
trainer.train(save_model_ckpt = False)
model.get_all_embeddings_and_nll(adata)
```

The above code snippet will instantiate an scETM model object, train the model, infer the unnormalized cell topics mixture of `adata` and store them in `adata.obsm['delta']`. We can also access the gene and topic embeddings via `adata.varm['rho']` and `adata.uns['alpha']`.

**Transfer learning with scETM**. When transferring from a reference dataset to a target dataset, we operate on the genes common to both datasets. For cross-species transfer, the orthologous genes based on the Mouse Genome Informatics database[64,65] are considered as common genes. The trained scETM encoder can be directly applied to an unseen target dataset, as long as the genes in the target dataset are aligned to the genes in the reference dataset. In the main text, for example, we trained scETM on a reference dataset and evaluated the scETM-encoder on a target dataset in six transfer-learning tasks (Fig. 4).

**Pathway enrichment analysis**. To assess whether a topic is enriched in any known pathway, one common way is to test for Over Representation Analysis (ORA)[66]. However, ORA requires choosing a subset of genes (e.g., from differential expression analysis). While we could choose the top genes scored by each topic, it requires some arbitrary threshold to select those genes. To avoid thresholding genes, we employed Gene Set Enrichment Analysis (GSEA)[44]. GSEA calculates a running sum of enrichment scores (ES) by going down the gene list that is sorted in the decreasing order by their association statistic with a phenotype.

In our context, we treated the gene scores under each topic from the genes-by-topics matrix (i.e., $\beta$) as the association statistic. The ES for a gene set S is the maximum difference between P_hit(S,i) and P_miss(S,i), where P_hit(S,i) is the fraction of genes in S weighted by their topic scores up to gene index i in the sorted list and P_miss(S,i) is the fraction of genes not in S weighted by their topic scores up to gene index i in the sorted list. The enrichment $p$-value for each gene set is computed by permutation tests by randomly shuffling the gene symbols on the sorted list (while keeping the gene-topic scores in the decreasing order) 1000 times to compute the null distribution of the ES for each gene set and each topic. The empirical $p$-value was calculated as $(N'+1)/(N+1)$, where $N'$ is the number of permutation trials in which ES is greater than the observed ES, and $N$ is the total number of trials (i.e., 1000). We then corrected the $p$-values for multiple testing using Benjamini–Hochberg (BH) method[67].

For AD and HP datasets, we used the MSigDB Canonical Pathways gene sets[68] as the gene set database in GSEA; and for MDD, we used PsyGeNET database[69] in order to find psychiatric disease-specific associations. We also run GSEA for scVI-LD gene loadings for comparison. The detailed pathway enrichment statistic can be found in Supplementary Data 1.

**Differential analysis of topic expression**. We aimed to identify topics that are differentially associated with known cell type labels or disease conditions. For topic

k and cell label j (i.e., cell type or disease condition), we first calculated the difference of the average topic activities between the cells with label j and the cells without label j. For each permutation trial, we randomly shuffled the label assignments among cells and recalculated the difference of average topic activities from the resulting permutation. The empirical $p$-value was calculated as $(N'+1)/(N+1)$, where $N'$ is the number of permutation trials in which the difference is greater than the observed difference, and $N$ is the total number of trials. To account for multiple hypotheses, we applied Bonferroni correction by multiplying the $p$-value by the product of the topic number and the number of labels. We performed $N = 100,000$ permutations.

We determined a topic to be differentially expressed (DE) if the Bonferroni-corrected $q$-value is lower than 0.01 and the mean difference is greater than 2 for cell-type DE topcis or 0.2 for disease DE topics. Supplementary Table 18 summarizes the number of DE topics we identified for each cell type and disease conditions from the AD and MDD data. We use the PanglaoDB database[70] to find the overlap between top genes of cell-type-specific DE topics and known cell type markers.

**Incorporation of pathway knowledge into the gene embeddings in p-scETM**. We downloaded the pathDIP4 pathway database from[48], and the Gene Ontology (Biological Processes) (GO-BP) dataset from MSigDB v7.2 Release[68]. Pathway gene sets or GO-BP terms containing fewer than five genes were removed. We represented the pathway knowledge as a pathways-by-genes $\rho$ matrix, where $\rho_{ij} = 1$ if gene set i contains gene j, and $\rho_{ij} = 0$ otherwise. We standardized each column (i.e., gene) of this matrix for numerical stability. During training the p-scETM, we fixed the gene embedding matrix $\rho$ to the pathways-by-genes matrix.

**Clustering performance benchmark and visualization**. We assessed the performance of each method by three metrics: Adjusted Rand Index (ARI)[71], Normalized Mutual Information (NMI) and k-nearest-neighbor Batch-Effect Test (kBET)[18]. ARI and NMI are widely-used representatives of two families of clustering agreement measures, pair-counting and information theoretic measures, respectively. A high ARI or NMI indicates a high degree of agreement for a given clustering result against the ground-truth cell type labels. We calculated ARI and NMI using the Python library scikit-learn[72].

kBET measures how well mixed the batches are based on the local batch label distribution in randomly sampled nearest-neighbor cells compared against the global batch label distribution. Average silhouette width (ASW)[40] indicates clustering quality using cell type labels. Silhouette width (SW) of a cell $i$ is the distance of cell $i$ from all of the cells within the same cluster subtracted by the distance of cell $i$ from cells in a nearest but different cluster, normalized by the maximum of these two values. ASW is the averaged SW over all the cells in a dataset and larger values indicate better clustering quality. Therefore, larger ASW indicates the higher distances between cell types and lower distance within cell type in the cell embedding space. We choose the distance function to be the euclidean distance. We adapted the Pegasus implementation[73] for kBET calculation, and set the $k$ to 15.

All embedding plots were generated using the Python scanpy package[16]. We use UMAP[39] to reduce the dimension of the embeddings to 2 for visualization, and Leiden[74] to cluster cells by their cell embeddings produced by each method in comparison. During the clustering, we tried multiple resolution values and reported the result with the highest ARI for each method.

For reproducibility, the evaluation and the plotting steps were implemented in a single `evaluate` function in the scETM package, which takes in an `AnnData` object with cell embeddings and returns a `Figure` object for the ARI, NMI, kBET, and embedding plot. For consistency, we used this function to evaluate all methods, including those written in R, where we used the `reticulate` package[75] to call our `evaluate` function.

We ran all methods under their recommended pipeline settings (Supplementary Methods), and we use batch correction option whenever applicable to account for batch effects. All results are obtained on a compute cluster with Intel Gold 6148 Skylake CPUs and Nvidia V100 GPUs. We limit each experiment to use 8 CPU cores, 192 GB RAM and 1 GPU.

**Efficiency and scalability benchmark of the existing methods**. To create a benchmark dataset for evaluating the run-time of each method, we merged MDD and AD, keeping the genes that appear in both datasets. We then selected the 3000 most variable genes using scanpy's `highly_variable_genes(n_top_genes=3000, flavor='seurat_v3')` function, and randomly sampled 28,000, 14,000, 70,000, and 148,247 (all) cells to create our benchmark datasets. The memory requirements reported in Fig. 3 were obtained by reading the `rss` attribute of the named tuple returned by calling `Process().memory_info()` from the `psutil` Python package[76]. For methods based on R, we use the `reticulate` package[75] to call the above Python function for consistency. We used the same settings (RAM size, number of GPUs, etc) as described in the Clustering performance benchmark and visualization section throughout the experiments.

**Reporting summary**. Further information on research design is available in the Nature Research Reporting Summary linked to this article.

## Data availability

The datasets analyzed during the current study are from publicly available repositories or data portals. The acquisition and quality control steps for all datasets are included in the supplementary information. The Human pancreatic islet dataset used in this study are available in the GEO or EMBL-EBI database under the accession codes GSE81076, GSE85241, GSE86469, E-MTAB-5061, and GSE84133. Mouse Pancreatic Islet dataset is available in the GEO database under the accession code GSE84133. Major Depressive Disorder dataset is available in the GEO database under the accession code GSE144136. Mouse Retina dataset is available in the GEO database under the accession codes GSE63473 and GSE81904. The Alzheimer's disease (AD) is available in Synapse under the access code syn18485175. The Tabula Muris dataset is available in the FigShare database under the accession code 27733. The Allen Brain Atlas datasets[77] namely human primary motor cortex and the primary motor cortex datasets and are available from the Allen Brain Portal [https://portal.brain-map.org/atlases-and-data/rnaseq/human-m1-10x, https://portal.brain-map.org/atlases-and-data/rnaseq/mouse-whole-cortex-and-hippocampus-10x, respectively]. Supplementary Information Section 1.1 describes the details in preprocessing these datasets.

## Code availability

scETM source codes as well as the benchmarking workflows have been deposited at the GitHub repository[78] (https://www.github.com/hui2000ji/scETM).

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

## Acknowledgements

We thank the current and former members of Li Lab, including Swapna Seshadri, Sydney Sue, Yue Lyu and Mojtaba Bahrami, who provided helpful comments in the early model development stage. Y.L. is supported by New Frontier Research Fund— Exploration (NFRFE-2019-00980) and Canada First Research Excellence Fund Healthy Brains for Healthy Life (HBHL) initiative New Investigator start-up award (G249591). Y.Z. is supported by Jacqueline Johnson Desoer Science Undergraduate Research Award (SURA). We also thank the reviewers for their constructive feedback.

## Author contributions

Y.L. and J.T. conceived of the study. Y.Z., H.C., Y.L. analyzed and interpreted the data, wrote the manuscript, and wrote the code for scETM. H.C. optimized and completed the final scETM code. Z.Z. ran some initial experiments. Y.L. and J.T. supervised the project. All authors approved the final manuscript.

## Competing interests

The authors declare no competing interests.
