## [Peer Review File · Nature Communications]

Learning interpretable cellular and gene signature embeddings from single-cell transcriptomic dataReviewers' Comments:

Reviewer #1:

Remarks to the Author:

The authors present a performant and interpretable VAE-based topic modeling framework adapted from the ETM model of Dieng et al. from the Blei lab and applied it to scRNA-seq datasets with various benchmarks. I believe that adapting ETM to scRNA-seq data with a batch correction modification is a great contribution for the community. Furthermore, literature review and the comparisons with other tools are also remarkably done. However, there are a few major issues on how the applications are presented. My comments and suggestions about how to improve the paper further can be found below.

Background:

- Page 2 second paragraph: It should be emphasized that the cellular programs that are shared across cell types can be as valuable as the exclusive programs typically detected via DE analysis. Approaches like topic modeling and NMF are highly important to reveal these programs. This is why such methods are increasingly being used in the scRNA-seq literature.
- One missed citation is the celda (<https://www.biorxiv.org/content/10.1101/2020.11.16.373274v1.full>) method by Campbell lab.
- Another one is scGen by Lotfollahi et al. which is a VAE for integrating datasets.
- Batch effects are modeled as intercepts so it's not clear to me what "batch effect embeddings" term on Page 3 means.

Overview figure:

- Figure 1 summarizes the mathematical aspects of the model, this might be good for an arxiv paper but is not intuitive and attractive for the general audience of Nature Communications. I think there must be a conceptual and visual way to explain the model. Please have a look at the first figures of the scVI, DCA and scalign papers, or Figure 5 in scGen paper (by Lotfollahi et al.).
- Plotting panel A in proper plate notation would improve interpretation for people who are familiar with the plate notation.
- Describing that 1..S are batches (or other unwanted effects) in panel B would improve the interpretation. Also, showing the dimensions of matrices would make it easier to understand.
- Plotting panel C similar to scVI with neurons, topics and genes and showing the conceptual topic/cell/gene embeddings e.g. distance of a cell to topics is its "topic loading" and gene to topics is topics' gene loading would greatly enhance the figure, I think.
- The merits and novelty of the method is not emphasized enough in this figure. It should be introduced as an integration method with useful extra features like transfer learning, gene and topic embeddings (in the context of gene-gene relationships and regulatory networks). One can add the cartoon version of Figure S2 as a panel here to demonstrate integration.

Integration and clustering:

- Adding two more rows to Table 1 would improve it: 1) ARI/NMI on the "uncorrected" data 2) name of the unwanted effect below the dataset name (e.g. seq type, individual etc).
- ARI relies on cell type labels which might be missing or incorrect. I would suggest using a metric of "mixedness" to also report how mixed the batch variable is in the detected clusters, in addition to NMI/ARI. LISI (in the Harmony paper) or kBet (Buettner et al.) are the metrics used for batch effect quantification (also see <https://genomebiology.biomedcentral.com/articles/10.1186/s13059-019-1850-9>) in all tables where ARI/NMI is used, especially Table 1.
- kBet citation refers to its preprint although the method was published two years ago, please update.
- How PCA panels in Fig S2 are processed is not described in the methods. I wonder if there was

something wrong in preprocessing because the structure of the data looks too different than the batch corrected data.

- Although integration is a major source of novelty, there is no main figure about it. I suggest moving one dataset from Fig S2 to a new "Figure 2" in a 4 x 4 format.
- The name of the section is clustering, however the authors actually evaluate the integration performance of the method USING clustering metrics. Considering that integration is a highlight, I suggest renaming clustering section to data integration.
- In more complex datasets there might be multiple categorical batch variables and/or numerical variables. It would be great if the authors comment on this, since it's relatively easy to extend the method to take multiple variables into account.
- In all batch correction models, I think it is important to see how the model handles cases where batches and meaningful biology overlap. E.g. One can keep one cell type only in one batch and see if batch correction removes the biological variation. This is a very important test about overcorrection, although there may not be ideal solution for such cases. I think neural networks can be more permissive and can perform better in such cases. I would be great to show it in comparison to other approaches.

Hyperparameter selection

- In order to guide the users who want to use the method with the right hyperparameters but without the cell type annotations, held-out log likelihoods (i.e. validation loss) as a more "native" metric can be utilized. Reporting these values for tables S2 and S3 would be very informative. I think ARI/NMI are overused in this study overall. Although clustering is important, it is by no means the ultimate metric to evaluate the performance of a scRNA-seq method.

Scalability

- Showing CPU and GPU runs of neural network-based models as two separate models in this plot would be informative for people who are curious about the speedup GPUs provide.

Transfer learning:

- This is a very important aspect of the model but the example doesn't fully show the strength of the model. Is the immune cell clustering observation reproducible across multiple runs? Cross-species analysis and classification are much more interesting tasks and can be strengthened further using the brain datasets from Allen Institute (<https://portal.brain-map.org/atlas-and-data/rnaseq>)
- The authors should add scVI, scVAE, scVI-LD results in the supplement.
- The authors should add a confusion matrix of the kNN classifier in the supplement.
- Please cite scArches <https://www.biorxiv.org/content/10.1101/2020.07.16.205997v1>.

Gene set enrichments:

- It is important to plot top N pathways and enrichment p-values given in Table S4,S5,S6. The readers should know how relevant top pathways are.
- Repeating model fits and gene enrichments only with ~20k protein-coding genes on standard chromosomes might improve interpretability considerably. Right now, there is a lot of AC, AL and LINC genes in Fig S1.
- Figure 4 Panel A: highlight the genes that are involved in the pathway written underneath the heatmap
- Although gene and topic embeddings are an extremely cool feature of the method, they are not shown at all. I suggest plotting gene and topic embeddings for some topics as in Dieng et al paper.
- It is not clear why the color scale in Panel C has so extreme values, compared to the [-4,6] range in Panel B. Setting the maximum values for zscores makes it hard to see smaller differences. Could the

authors comment on that?

p-scETM:

- Adding a conceptual plot in Figure 5 explaining how rho matrix is prepared and how the overall model looks like would help the readers.
- As a popular gene set resource, please consider providing the GO version of p-scETM.

Code:

- Please cite PyTorch (<https://github.com/pytorch/pytorch/blob/master/CITATION>) and give credit to Adji Dieng for the implementation if your implementation is not from scratch.
- Unfortunately the code is not available, please make it available.
- Especially for the scRNA-seq tools implemented in Python, integration with scanpy, as the most popular framework in the field, is critical. It would be great to have a function e.g. `scetm.tl.infer(adata)` which stores topic and gene embeddings as well as the loadings in `.obs` and `.varm` fields of an AnnData object.
- Please describe which package is used for ARI and NMI and cite if necessary.
- Consider using the NB distribution when scVI is applied to UMI datasets, since UMI data is not zero inflated, see Suppl. Figure 1 in the DCA paper.

Conceptual questions:

- How are "stop words" (e.g. MALAT1) handled? Do you see ubiquitously expressed genes dominating all topics?
- How is the variation in library sizes (see the `L_n` variable in scVI paper) handled?
- Is the encoder input log transformed and/or TP10K normalized?

Minor:

- I suggest supporting this statement with a citation or remove: "Variational autoencoders (VAE) [26] is an efficient probabilistic framework known to better account for noise compared to conventional autoencoders."
- The authors use read count matrix and read counts throughout the manuscript however some datasets use UMI counts and not read counts. I suggest removing "reads" and just using "counts" to imply that it can be read or UMI counts.

Best regards,
Gokcen Eraslan.

Reviewer #2:

Remarks to the Author:

In this work, the authors propose single-cell Embedded Topic Model (scETM) to obtain interpretable embeddings of single cell data matrix. They mention that scETM is more scalable, obtain high interpretability and is able to transfer knowledge across datasets. Also they show how pathway information could be used in the learning step.

Strengths of the work:

- i. The paper is well-written and easy to follow. The authors provide conditions in which they ran the baselines, provided significance values in hypothesis testing scenarios and compared multiple state-of-the-art methods on their datasets.

- ii. Transferring the trained model on Human genes to mouse genes still obtained good performance (while competing methods suffer) is an interesting finding.
- iii. Fixing the pathway matrix and then obtaining interpretable topics is also an interesting finding.

Weaknesses:

- i. Lack of novelty: Deep learning methods for clustering single cells is not new in the literature. In fact, the proposed approach is very similar in spirit to scVI where the authors also used a VAE. scVI-LD also uses a linear decoder for interpretability just as proposed here. The matrix factorization approach to single cell clustering and dimension reduction has been proposed in prior published work [1]. Considering all these, I find the contribution of the current work limited.
- ii. The baseline/competing methods are run with their default settings, while the proposed scETM could leverage different hyperparameters. While the authors do show a robustness to hyperparameters table, it would be nice to compare baselines on datasets which they used in their publication (Example - strongly recommend showing results of scETM on either of CORTEX or HEMATO datasets made publicly available by scVI authors). This would lead to a more direct comparison. Also, it would be nice to see what scVI clustering metrics look like for latent dimension of 100 (equivalent to 100 topics used by scETM). The current latent dimension of 10 seem too small for scVI.
- iii. It is not clear how scETM is better with regards to interpretability for the range of experiments conducted. There are no metrics for interpretability. What are the interpretable clusters obtained from an approach such as scVI-LD on AD or MDD datasets?
- iv. It is not clear to me what the false discovery of significant genes is in topics associated with cell types. For example, Page 6 Last paragraph "In AD, the top 30 genes from topic 75 are highly enriched in amyloid fiber formation". Are all the 30 genes related to amyloid fiber formation in the biology literature? Or are there any false positives? Similarly for the statement, "In MDD, topic 7 is enriched for neurodegenerative diseases such as Parkinson's , ...". In this line, there is not even a mention of how many top genes in the topic led to this claim.

Additional comments:

1. Page 5 Last paragraph "We trained an scETM model on TM-FACS and .. evaluated it using the MP data, which only contains mouse pancreatic islet cells". Does the TM-FACS dataset not contain any mouse pancreatic islet cells?
2. Page 5 Section Scalability. What aspect of the proposed model led to run-time gain is not clear. Stochastic variational inference, minibatch parameter update is applicable to other deep learning models such as scVI also. What aspect of scETM is unique for scalability, which cannot be applied for other deep learning methods?
3. Page 8 Section Pathway-informed scETM Topics. "6 topics related to nutrient digestion and metabolism", "we found 8 topics with top pathways known as therapeutic targets for MDD treatment". How would you group together topics in a setting where ground truth pathway knowledge is not available? The number of topics for a group seems to vary. Not clear how to group together topics without labeled/annotated information.

Reproducibility:

I could not access the link to code (<https://github.com/li-lab-mcgill/scETM>) provided by the authors. Hence unable to ensure whether it is easy to reproduce the work.

Minor typo:

Page 7 Last paragraph Line 6 "learn only the pathways by topics embedding" should be "learn only the topics by pathways embedding".

References:

[1] Mukherjee S, Zhang Y, Fan J, Seelig G, Kannan S. Scalable preprocessing for sparse scRNA-seq data exploiting prior knowledge. *Bioinformatics*. 2018 Jul 1;34(13):i124-32.

Point-by-Point Response to Reviewers

Overview

We would like to thank both Reviewers for the thoughtful and constructive comments. We have made substantial amounts of effort to address every single comment.

We highlight a few:

1. **Highlighting the strengths of our method:** We demonstrated and highlighted the strengths of our methods in terms of batch effect correction (**Response 10 to Reviewer 1**), transferability, interpretabilities in the writings, presentations (e.g., the new Figure 1 (**Response 5 to Reviewer 1**), visualizing gene embeddings and topic embeddings in **Response 27 to Reviewer 1**), and a new set of comprehensive experiments comparing state-of-the-art (SOTA) methods;
2. **Batch overcorrection analysis (Response 17 to Reviewer 1):** We performed batch over-correction analysis comparing scETM with 7 SOTA methods using two separate datasets each with disproportional cell types over each batch;
3. **Transfer learning (Response 20 and 21 to Reviewer 1; Response 6 to Reviewer 2):** We conducted more comprehensive cross-tissue, cross-species transfer-learning with added human and mouse Allen Brain datasets and obtained remarkable results;
4. **Interpretability comparison (Response 4 to Reviewer 2):** We quantified the interpretability by the enrichments of known pathways in comparison with scVI-LD;
5. **Code:** We made our code publicly available via Github (<https://www.github.com/hui2000ji/scETM>) with easy-to-follow tutorials and also easy-to-use API well integrated with the scanpy and PyPI ecosystem.

The revised contents are highlighted in **Dark Red** in the main text. For easy reference within the rebuttal letter, we included some of the figures and tables in the letter and referred to them by the index number of our response. For example, when responding to Reviewer 1 comment 5, we used **Figure R5** (the new Figure 1 in the main text) in the place of the response.

Although we tried our best to make sure that the figures and tables in this rebuttal letter match with those in our revision, in the case where there are some occasional small discrepancies between the ones in this letter and the revision (e.g., some revised text, figure legends, reported performances), please refer to the corresponding content in the revision as our latest version.

Reviewer #1 (Expertise: Deep learning for scSeq data analysis):

The authors present a performant and interpretable VAE-based topic modeling framework adapted from the ETM model of Dieng et al. from the Blei lab and applied it to scRNA-seq datasets with various benchmarks. I believe that adapting ETM to scRNA-seq data with a batch correction modification is a great contribution for the community. Furthermore, literature review and the comparisons with other tools are also remarkably done. However, there are a few major issues on how the applications are presented. My comments and suggestions about how to improve the paper further can be found below.

Background:

- Page 2 second paragraph: It should be emphasized that the cellular programs that are shared across cell types can be as valuable as the exclusive programs typically detected via DE analysis. Approaches like topic modeling and NMF are highly important to reveal these programs. This is why such methods are increasingly being used in the scRNA-seq literature.

#1 Our response:

Thank you for the suggestion. We revised the paragraph as follows (also highlighted in Background section):

Latent topic models are a popular approach in mining genomic and healthcare data [21-23] and are increasingly being used in the scRNA-seq literature [24]. Specifically, in topic modeling, we infer the topic distribution for both the samples and genomic features by decomposing the samples-by-features matrix into samples-by-topics and topics-by-features matrices, which also be viewed as a probabilistic non-negative factorization (NMF). Importantly, the top genes under each latent topic can reveal the gene signatures for specific cellular programs, which can be shared across cell types or exclusive to a particular cell type. Traditionally, the latter are detected via differential expression analysis at individual gene levels, which has limited statistical power in scRNA-seq data analysis because of the sparse gene counts, small number of unique biological samples, and the burdens of multiple testings.

- One missed citation is the celda (<https://www.biorxiv.org/content/10.1101/2020.11.16.373274v1.full>) method by Campbell lab.

#2 Our response:

Thank you for the suggestion. We cited this paper as reference [24] in the above interpretability paragraph in the **Background** section:

“Latent topic models are a popular approach in mining genomic and healthcare data [21–23] and are increasingly being used in the scRNA-seq literature [24].”

- Another one is scGen by Lotfollahi et al. which is a VAE for integrating datasets.

#3 Our response:

Thank you. We cited this paper as reference [26] when introducing transferability and deep learning in the Background section:

“As the number and size of scRNA-seq datasets continue to increase, there is an increasingly high demand for efficient exploitation and knowledge transfer from the existing reference datasets [25,26].

...

Deep learning approaches, especially autoencoders, have demonstrated promising performance in scRNA-seq data modeling ... Lotfollahi et al. developed a VAE model called scGen to infer the expression difference due to perturbation conditions by latent space interpolation [26].”

- Batch effects are modeled as intercepts so it's not clear to me what "batch effect embeddings" term on Page 3 means.

#4 Our response:

For better terminology, we changed it to “batch-effect linear intercepts”. Please see the second last paragraph in Background section:

“In this paper, we present single-cell Embedded Topic Model (scETM) ... The scETM simultaneously learns the encoder network parameters and a set of highly interpretable gene embeddings, topic embeddings, and batch-effect linear intercepts from scRNA-seq data.”

Overview figure:

- Figure 1 summarizes the mathematical aspects of the model, this might be good for an arxiv paper but is not intuitive and attractive for the general audience of Nature Communications. I think there must be a conceptual and visual way to explain the model. Please have a look at the first figures of the scVI, DCA and scalign papers, or Figure 5 in scGen paper (by Lotfollahi et al.).

#5 Our response:

Thank you for this suggestion. We have completely redrawn Figure 1 (**Figure R5**) to capture the algorithmic essence and the applications of our scETM model in efforts to make it accessible to the general audience of *Nature Communications*.

Fig. 1

(a) scETM modeling of single-cell transcriptomes across multiple experiments or studies

(b) Transfer learning to cluster cells from unseen data

(c) Genes and topics embedding cluster

Figure R5 (Figure 1 in the main text). scETM model overview. Given as input the scRNA-seq data matrices across multiple experiments or studies (i.e., batches), scETM models the single-cell transcriptomes using an embedded topic modeling approach. (a) Each scRNA-seq profile serves as an input to a variational encoder (VAE) as the normalized gene counts. The VAE produces a stochastic sample of the latent topic mixture ($\theta_{s,d}$ for batch $s = 1 \dots S$ and cell $d = 1 \dots N_s$), which can be used for clustering cells (see panel b). The linear decoder learns topic embedding and gene embedding, which can be used to analyze cellular programs via enrichment analyses (see panel c). (b) Workflow used to perform transfer learning. The trained scETM-encoder on a reference scRNA-seq dataset is used to learn the cell topic mixture θ^* from an unseen scRNA-seq dataset *without training* them. The resulting cell mixtures are then visualized via UMAP visualization. (c) Exploring gene embeddings and topic embeddings. Because the genes and topics share the same embedding space, we can explore their connections via UMAP visualization or annotate each topic via enrichment analyses using known biological processes.

- Plotting panel A in proper plate notation would improve interpretation for people who are familiar with the plate notation.

#6 Our response:

Agreed. We have revised the panel A with proper plate notation and placed it into supplementary information **Figure S1**. Please see **Figure R6 (a)** below.

Figure R6 (Supplementary Figure S1 in Supplementary Information). scETM model details. (a) The plate model for scETM. We model the scRNA-profile count matrix $y_{d,g}$ in cell d and gene g across S batches by a multinomial distribution with the rate parameterized by cell topic mixture θ , topic embedding α , gene embedding ρ , and batch effects λ . (b) Matrix factorization view of scETM. (c) Encoder architecture for learning the cell topic mixture θ .

- Describing that 1..S are batches (or other unwanted effects) in panel B would improve the interpretation. Also, showing the dimensions of matrices would make it easier to understand.

#7 Our response:

Agreed. Please see **Figure R6 (b)** above or the new **Figure S1 (b)**.

- Plotting panel C similar to scVI with neurons, topics and genes and showing the conceptual topic/cell/gene embeddings e.g. distance of a cell to topics is its “topic loading” and gene to topics is topics' gene loading would greatly enhance the figure, I think.

#8 Our response:

Agreed. Please see **Figure R5** or the new **Figure 1 (a)**.

- The merits and novelty of the method is not emphasized enough in this figure. It should be introduced as an integration method with useful extra features like transfer learning, gene and topic embeddings (in the context of gene-gene relationships and regulatory networks). One can add the cartoon version of Figure S2 as a panel here to demonstrate integration.

#9 Our response:

Thank you for this suggestion. Please see **Figure R5** or the new **Figure 1 (b) and (c)**.

Integration and clustering:

- Adding two more rows to Table 1 would improve it: 1) ARI/NMI on the “uncorrected” data 2) name of the unwanted effect below the dataset name (e.g. seq type, individual etc).

#10 Our response:

We have added the following to **Table 1** (ARI table):

	MP	HP	TM	AD	MDD
scETM	0.951	0.943	0.761	0.996	0.717
scETM- λ	0.851	0.474	0.629	0.996	0.719
Batch Effect	Strain	Technology	Technology	Individual	Individual

We have added the following to **Table S1** (NMI table):

	MP	HP	TM	AD	MDD
scETM	0.902	0.896	0.856	0.987	0.620
scETM- λ	0.819	0.644	0.819	0.998	0.604
Batch Effect	Strain	Technology	Technology	Individual	Individual

Here “scETM” indicates scETM performance with batch correction and “scETM- λ ” indicates scETM performance without batch correction.

The “Batch Effect” row lists the names of the unwanted effects for each dataset.

Compared to scETM- λ without batch correction λ , scETM confers higher ARI in 3 out of the 5 datasets and higher NMI in 4 out of the 5 datasets. Improvement over the Human Pancreas (HP) dataset is remarkably high, implying an effective correction of the confounder due to the scRNA-seq technology differences. We observe no improvement in the AD dataset in terms of both ARI and NMI and small improvement in MDD only in terms of NMI. This implies a lesser

concern of batch effects from the individual brain sample donors, which were used as the batches in fitting the batch effect λ in scETM.

We added the above description to the **Data Integration** section of the **Results**.

- ARI relies on cell type labels which might be missing or incorrect. I would suggest using a metric of "mixedness" to also report how mixed the batch variable is in the detected clusters, in addition to NMI/ARI. LISI (in the Harmony paper) or kBET (Buettner et al.) are the metrics used for batch effect quantification (also see <https://genomebiology.biomedcentral.com/articles/10.1186/s13059-019-1850-9>) in all tables where ARI/NMI is used, especially Table 1.

#11 Our response:

Thanks for your suggestion. We have now evaluated the batch mixing aspect of scETM and other methods using kBET (**Table 2**). Additionally, we examined to what extent scETM's batch mixing performance can be improved by introducing an adversarial loss term to scETM (**Methods scETM plus adversarial loss for further batch correction**). Briefly, we use a discriminator network (a two-layer feedforward network) to predict batch labels using the cell topic mixture embedding generated by the encoder network. We optimize both networks in an alternative way: (1) we optimize the encoder network by minimizing the negative ELBO plus the negative batch-prediction cross-entropy loss from the discriminator, i.e. the encoder will try to fool the discriminator; (2) we optimize the discriminator network by minimizing the cross-entropy loss. We observe notable improvement on kBET with similar ARI and NMI scores at the cost of up to 50% more running time (**Table 1**, row "scETM + adv."). This shows how *expandable* our scETM can be. For the subsequent analyses, we opted to use the results from scETM (without the adversarial loss but with the linear batch correction λ) because of its simpler design, scalability, comparable ARI scores, and less aggressive batch correction (please see our **Response #17** on this point).

Meanwhile, we also see that some methods conferred competitive kBET (**Table 2**) but low ARI (**Table 1**), suggesting potential overcorrection of batch effects. Specifically, while LIGER conferred the highest kBET in our analysis, its ARI is lower than several state-of-the-art methods including our scETM (without the adversarial batch correction loss). This is consistent with the observation made by Luecken et al. (bioRxiv 2021), who reported that, while LIGER is among the top batch mixing methods, it often removes biological variations as a tradeoff.

Please also kindly refer to our **Response #17** for more detailed analysis of overcorrection. In short, both the ARI and kBET need to be considered together when evaluating how well the method captures the meaningful transcriptional programs while accounting for batch effects.

Reference:

Luecken, M., Büttner, M., Chaichoompu, K., Danese, A., Interlandi, M., Mueller, M., Strobl, D., Zappia, L., Dugas, M., Colomé-Tatché, M., & Theis, F. (2020). Benchmarking atlas-level data

integration in single-cell genomics. BioRxiv, 2020.05.22.111161.
<https://doi.org/10.1101/2020.05.22.111161>

- kBET citation refers to its preprint although the method was published two years ago, please update.

#12 Our response:

Thank you for noticing that. We have updated the kBET citation:

Maren Büttner, Zhichao Miao, F Alexander Wolf, Sarah A Teichmann, and Fabian J Theis. A test metric for assessing single-cell rna-seq batch correction. *Nature methods*, 16(1):43–49, 2019

- How PCA panels in Fig S2 are processed is not described in the methods. I wonder if there was something wrong in preprocessing because the structure of the data looks too different than the batch corrected data.

#13 Our response:

The PCA embeddings were obtained by applying PCA to the *raw* data, without preprocessing. Specifically, we used the scanpy package to calculate PCA embeddings of raw data with the number of PCs set to 50. No preprocessing step was applied before PCA. However, since PCA is not a state-of-the-art method for scRNA-seq modeling, we decided to remove the PCA UMAP plots in this revision.

- Although integration is a major source of novelty, there is no main figure about it. I suggest moving one dataset from Fig S2 to a new "Figure 2" in a 4 x 4 format.

#14 Our response:

Thank you for this suggestion. We have now added the 4 x 4 figure using the Mouse Retina (MR) dataset in **Figure R14** (also the new **Figure 2** in the main text) to illustrate integration and batch (over-)correction by each method. We obtained the MR dataset from <https://hemberg-lab.github.io/scRNA.seq.datasets/mouse/retina/>, which is a collection of MR scRNA-seq datasets from two separate studies, namely Macosko et al., (*Cell* 2015) and Shekhar et al., (*Cell*, 2016). Here we consider the two different source studies as two batches, named from now on the Macosko batch and Shekhar batch.

In line with your insightful comment regarding the batch overcorrection (please see additional description in our **Response 17** below), we chose to show the UMAP visualization of this dataset because of the uneven distribution of some cell types between the two studies in order to illustrate the performance difference among the methods in comparison.

- The name of the section is clustering, however the authors actually evaluate the integration performance of the method USING clustering metrics. Considering that integration is a highlight, I suggest renaming clustering section to data integration.

#15 Our response:

Agreed. We changed the section name from “Clustering” to “Data integration”.

- In more complex datasets there might be multiple categorical batch variables and/or numerical variables. It would be great if the authors comment on this, since it's relatively easy to extend the method to take multiple variables into account.

#16 Our response:

We agree with the reviewer that the ability to take multiple categorical batch variables into account can be useful in complex datasets. Our comment is as follows and has been added to the **Discussion section**:

Here we only consider a single batch effect correction of single categorical batch variables. As a future work, we will extend scETM to correct for multiple categorical batch variables. For a small number of categorical batch variables, we may use several sets of batch intercept terms to model them. For hierarchical batch variables, we may use a tree of batch intercept terms. For numerical batch effects such as age, convert them into categorical variables by numerical ranges. When the number of batch variables becomes larger, we consider three strategies. First, we can add the batch variables as the covariates in the linear regression on the gene expression and fit the linear coefficients each corresponding to a sample-dependent batch variable. Second, we can factorize the batches-by-genes into batches-by-factors and factors-by-genes. Learning the two matrices will be similar to the ETM algorithm. Third, we can extend our current scETM+adv to correct for both categorical and continuous batch variables with a discriminator network, which predicts batch effects using the encoder-generated cell topic mixture.

- In all batch correction models, I think it is important to see how the model handles cases where batches and meaningful biology overlap. E.g. One can keep one cell type only in one batch and see if batch correction removes the biological variation. This is a very important test about overcorrection, although there may not be ideal solution for such cases. I think neural networks can be more permissive and can perform better in such cases. It would be great to show it in comparison to other approaches.

#17 Our response:

Thank you for this great suggestion. We tested the extent of overcorrection by our method as well as the existing methods using two datasets, namely the Human Pancreas (HP) dataset and the Mouse Retina (MR) dataset (Macosko & Shekhar datasets obtained from <https://hemberg-lab.github.io/scRNA.seq.datasets/mouse/retina/>).

For the HP data, we manually removed *beta* cells from all 5 batches except for CelSeq2, resulting in the cell type distributions shown in the table below. We expect that, if a method is guilty of batch effect overcorrection, it would assign beta cells to other non-beta cell clusters in the latent space by forcing the alignment of different batches. Consequently, such methods would have poor clustering scores.

	CelSeq	CelSeq2	Fludigm C1	InDrop	SmartSeq2
acinar	229	274	21	1152	188
activated_stellate	19	90	16	294	55
alpha	213	844	241	2309	1008
beta	0	445	0	0	0
delta	50	203	25	608	127
ductal	304	257	34	915	444
endothelial	5	21	14	235	21
epsilon	1	4	1	16	8
gamma	18	110	18	266	213
macrophage	1	15	1	55	7
mast	1	6	3	39	7
quiescent_stellate	1	12	1	160	6
schwann	1	4	5	13	2

We evaluated all of the methods on this dataset using 3 metrics: ARI, kBET, and average silhouette width (ASW) (Batool and Hennig, 2021)¹. Briefly, silhouette width (SW) of a cell *i* is the distance of cell *i* from all of the cells within the same cluster subtracted by the distance of cell *i* from cells in a nearest but different cluster, normalized by the maximum of these two values. ASW is the averaged SW over all the cells in a dataset and larger values indicate better clustering quality. We measured the overall ASW as well as the ASW for only the beta cells (i.e., ASW-beta). The results are summarized in the table below. As we can see, scETM strikes a good balance between discriminating cell types (ARI: 0.9265) and integrating different batches (kBET: 0.1247). Adding the adversarial loss to the scETM (i.e., scETM+adv) increased kBET from 0.1247 to 0.3445 (while maintaining ARI above 0.92) but greatly compromised ASW-beta,

¹ Batool, F. & Hennig, C. Clustering with the Average Silhouette Width. *Comput Stat Data An* 158, 107190 (2021).

suggesting a more aggressive overcorrection than the linear batch effect correction in sceTM. Similarly, Liger performed the best in kBET (0.5978) but conferred a much lower ARI (0.8476) and low ASW-beta (0.0912), indicating a severe overcorrection of the batch effects (i.e., mixing beta cells with other cells).

	ARI	ASW	ASW-beta	kBET
Seuratv3	0.9225	0.2732	0.1490	0.3337
Harmony	0.9024	0.2323	0.1558	0.2637
Scanorama	0.8989	0.3832	0.3203	0.1067
Liger	0.8476	0.1946	0.0912	0.5978
scVI-LD	0.5077	0.1397	0.6135	0.0031
scVI	0.6975	0.1579	0.4044	0.0567
scETM + adv.	0.9265	0.3026	0.0045	0.3445
scETM	0.9298	0.3525	0.5370	0.1247

We then visualize the clustering by UMAP (**Figure R17**) to examine how the beta cells are assigned to different clusters. We found that beta cells (colored in red) are clustered separately by scETM from other cell types. In contrast, beta cells are mixed up with other cell types by methods including Harmony and LIGER, which overcorrect the batch effects when integrating the five batches. Visually, we also observe that our scETM + adv method moves the beta cell cluster closer to the alpha cell cluster, suggesting a higher level of overcorrection compared to scETM.

For the MR dataset, many cell types are uniquely present in the Macosko batch as shown in the table below. Also, there is a large difference in the cell proportion between the two batches. In particular, rods are only 0.35% in Shekhar but 65% in Macosko. In this scenario, we expect that methods which overcorrect the batch effect would tend to mix rods with cells of other cell types from Shekhar batch, resulting in low ARI and high kBET. On the contrary, a desirable integration method would strike a balance between the ARI and kBET on this combined dataset. Therefore, this imposes a great challenge on the integration methods.

Overall, scETM achieves the highest ARI with 0.859 and modest kBET, suggesting its ability to capture the true biology from the data without over-correcting the batch effects. Given that it is impossible to achieve both high ARI and high kBET in this setup, scETM conferred a reasonable kBET score. Indeed, scETM's kBET is low mainly because of the fact that it leads to the separate clustering of astrocyte cells. In contrast, LIGER is more aggressive in the batch correction, resulting in the highest kBET score of 0.204 but lowest ARI score of 0.395. We further investigate the extent of improving kBET while maintaining a high ARI score by introducing an adversarial loss term to the scETM framework (scETM + adv.). Indeed, scETM + adv conferred a much improved kBET of 0.127 while a still reasonably high ARI score of 0.762.

With the additional adversarial training regime, scETM+adv conferred an notably increased kBET (0.1270) with a tradeoff on the ARI score (0.7619). Visualizing the clustering of each method using UMAP (**Figure R14**) confirms the quantitative clustering results.

	Shekhar et al.	Macosko et al.
amacrine	252	4426
astrocytes	0	54
bipolar	23494	6285
cones	48	1868
fibroblasts	0	85
ganglion	0	432
horizontal	0	252
microglia	0	67
muller	2945	1624
pericytes	0	63
rods	91	29400
vascular_endothelium	0	252

	ARI	ASW	kBET
Harmony	0.7632	0.1366	0.0631
Scanorama	0.7795	0.0934	0.0625
Seuratv3	0.7813	0.3464	0.0534
Liger	0.7135	0.2215	0.1761
scVI	0.7827	0.1331	0.0732
scVI-LD	0.7182	0.1964	0.0217
scETM + adv.	0.7720	0.2669	0.1410
scETM	0.8593	0.2873	0.0656

We added a section called **Batch overcorrection analysis** under Results to incorporate the above results in the revision.

Hyperparameter selection

- In order to guide the users who want to use the method with the right hyperparameters but without the cell type annotations, held-out log likelihoods (i.e. validation loss) as a more "native" metric can be utilized. Reporting these values for tables S2 and S3 would be very informative. I think ARI/NMI are overused in this study overall. Although clustering is important, it is by no means the ultimate metric to evaluate the performance of a scRNA-seq method.

#18 Our response:

Indeed, cluster agreement metrics are not the only metrics for evaluating scRNA-seq methods, and are not available to unannotated datasets. We agree that, without any cell label, the negative log-likelihood (NLL) on held-out samples is a principled way for model selection. We computed the held-out (10%) NLL and updated **Table S2** and summarize our observations as follows:

1. The best model chosen based on NLL has a similar ARI to the highest ARI that can be achieved.
2. scETM is robust to different architectures in terms of the NLL (**Table S2**).
3. ARI and NLL form modest negative correlation, implying an agreement between the two metrics (**Figure R18**)

We added this to the **Data Integration** section under Results.

Figure R18 (Figure S2 in main text). Relationship between average negative log-likelihood (NLL) and adjusted Rand Index (ARI) on the TM dataset. Each point denotes the performance of a trained scETM instance with a specific hyperparameter configuration (e.g., encoder hidden size, topic number, embedding dimensions, etc), averaged over three runs with different random seeds.

Scalability

- Showing CPU and GPU runs of neural network-based models as two separate models in this plot would be informative for people who are curious about the speedup GPUs provide.

#19 Our response:

Agreed. Although it has been widely accepted by the deep learning community that using GPUs for computing results in significant (~10x) speedup, the adoption of GPUs in the computational biology community is beginning to catch up. In our non-exhaustive experiment on the Mouse Pancreas dataset, training scETM for 1000 steps on the 6-core Core i7 10750H CPU requires 650 seconds, while on a RTX 2070 laptop GPU it only takes 50 seconds -- 13 times speedup over the CPU computer. With better GPUs such as the V100, the speedup will become more dramatic. We added this observation as the last paragraph to the **Scalability section** in the main text to highlight the benefits of using GPU computing.

Transfer learning:

- This is a very important aspect of the model but the example doesn't fully show the strength of the model. Is the immune cell clustering observation reproducible across multiple runs? Cross-species analysis and classification are much more interesting tasks and can be strengthened further using the brain datasets from Allen Institute (<https://portal.brain-map.org/atlas-and-data/rnaseq>)

#20 Our response:

Regarding reproducibility of immune cell clustering, we repeated the same experiment 3 times with different random seeds and observed consistently that B and T cells are close to each other and distant from macrophages (**Figure R20-1**).

Figure R20-1. Reproducibility of the separation of T-cells and B-cells from macrophages. We trained scETM on TM (FACS) and applied it to the Mouse Pancreas data. To assess the reproducibility, we repeated the experiments 3 times using different random seeds to initialize the model.

Thank you for the suggestion on the brain datasets from Allen Institute. We experimented scETM on the Human M1C - 10X Genomics (HumM1C), and the Mouse primary motor area (MusMOp) 10x Genomics (2020) data obtained from the Allen Brain map data portal (<https://portal.brain-map.org/atlasses-and-data/rnaseq>). The HumM1C dataset includes the profiles of single-cell transcriptomes in human primary motor cortex, and the MusMOp dataset include single-cell transcriptomes from primary motor area in mice. We chose to transfer between the human M1C and mouse MOp because of the high number of shared cell types between the brain regions of the two species. The batches for HumM1C are the two post-mortem human brain M1 specimens and the two mice for MusMOp.

We then performed a more comprehensive set of cross-tissue and cross-species transfer learning analysis:

1. Transfer between the Fluorescence-Activated Single Cell Sorting Tabula Muris (TM (FACS)), which profile single cell transcriptomes across multiple primary tissues including pancreas from mice, and the Mouse Pancreas dataset, which only assays scRNA-seq in mouse pancreas;
2. Transfer between the Human Pancreas (HP) dataset and the Mouse Pancreas (MP) dataset;
3. Transfer between the HumM1C dataset and the MusMOp dataset.

As a comparison, we evaluated and visualized the clustering results in all transfer learning using scETM, scVI-LD, and scVI (**Figure R20-2; Table R21**). Overall, we observe superior transfer learning performance from scETM compared to scVI and scVI-LD. In particular, scETM achieved the highest ARI across all transfer-learning tasks and competitive kBET scores. For instance, scETM trained on FACS-TM on heterogeneous tissues clusters quite well the MP cells (ARI: 0.941; kBET: 0.339). Remarkably, scETM trained only on the MP dataset can cluster reasonably well the FACS-TM single cells from diverse primary tissues including pancreas. This implies that scETM does not merely learn cell-type-specific signatures but also some underlying transcriptional programs that are generalizable to unseen tissues, suggesting its potential utility of annotating cells of unknown cell types.

Cross-species transfer experiments between HP and MP are also remarkably good, implying conserved pancreas functions that are better captured by scETM than by scVI and scVI-LD. On the other hand, cross-species transfer between MusMOp and HumM1C is a much more challenging task due to the evolutionarily divergent functions of the brains between the two species. Nonetheless, scETM conferred a much higher ARI of 0.696 for the MusMOp to HumM1C transfer-learning tasks and ARI 0.167 for the HumM1C to MusMOp task. In contrast,

scVI-LD and scVI did not work well on these tasks with ARI scores lower than 0.1. It is possible that both scVI-LD and scVI over-corrects batches since the kBET for both methods are relatively high. It is also possible that the tri-factorization gene embedding learning strategy and the topic model formalism implemented in scETM better capture the transferable cellular programs than the network decoder and standard bi-factorization embedding approaches from scVI and scVI-LD, respectively.

We added the above to the **Transfer learning** section of the **Results**.

- The authors should add scVI, scVAE, scVI-LD results in the supplement.

#21 Our response:

Thank you for this suggestion. We added the comparison with scVI and scVI-LD in terms of ARI and kBET as shown in **Table R21** (or **Supplementary Table S10**) and discussed it in the **Transfer learning** section.

ARI:

Type of transfer	Cross-tissue			Cross-species		Cross-species	
	src/tgt	MP -> TM (FACS)	TM (FACS) -> MP	TM (FACS) w/o Panc -> MP	MP -> HP (InDrop)	HP (InDrop) -> MP	MusMOp -> HumM1C
scVI	0.5075	0.4844	0.4197	0.5236	0.4251	0.0900	0.0252
scVI-LD	0.5159	0.3985	0.4317	0.6902	0.4757	0.0895	0.0375
scETM	0.5659	0.9409	0.8072	0.8680	0.7998	0.7105	0.3515

kBET:

Type of transfer	Cross-tissue			Cross-species		Cross-species	
	src/tgt	MP -> TM (FACS)	TM (FACS) -> MP	TM (FACS) w/o Panc -> MP	MP -> HP (InDrop)	HP (InDrop) -> MP	MusMOp -> HumM1C
scVI	0.0874	0.2570	0.2672	0.1707	0.2276	0.9002	0.8478
scVI-LD	0.0559	0.2564	0.2948	0.1425	0.2204	0.9181	0.8524
scETM	0.0585	0.3388	0.4772	0.1782	0.2930	0.7391	0.7775

Table R21. Evaluation of cross-tissue and cross-species transfer learning.

Abbreviations: MP: Mouse Pancreas; HP (InDrop): Human Pancreas sequenced with InDrop technology; TM (FACS): A subset of Tabula Muris sequenced with Fluorescence-Activated Single Cell Sorting; TM (FACS) w/o Panc: TM (FACS) without Pancreas; HumM1C: human primary motor cortex (from Allen Brain map); MusMOp: Mouse primary motor area (from Allen Brain map).

Note:

- TM (FACS) w/o Panc is a subset of TM (FACS) where all cells in the tissue “pancreas” are removed. This is to demonstrate that scETM can transfer knowledge to new tissues.
- HP (InDrop) is the subset of HP sequenced with InDrop, which is the same technology used to obtain MP. This is to prevent simultaneous transfer between species and technology, which might be too hard for the zero-shot transfer.
- The high kBET achieved by scVI and scVI-LD when transferring between MusMOp and HumM1C is attributed to the extremely low ARI, i.e. cell types are also mixed together.

Together from our extensive transfer learning experiments with seven dataset pairs, scETM demonstrates excellent capabilities in zero-shot transfer learning (i.e., the model is applied to the target dataset without being trained on it) across technology/tissue and species than scVI and scVI-LD. We did not compare with scVAE as it is much slower and often fails to converge (see **Table 1**).

- The authors should add a confusion matrix of the kNN classifier in the supplement.

#22 Our response:

Thank you for the suggestion. We felt that the kNN classification is tangential to our current study. Transferring a trained supervised learning model from reference to a target dataset requires aligning the cell type labels between the target dataset and the reference dataset. This does not work with our zero-shot transfer learning as demonstrated in this study. In our zero-shot transfer, the model is only trained on the reference dataset without knowing the cell type labels and then directly applied to the target dataset. Therefore, we wanted to sharpen the focus of our study by removing the kNN classifier results and focus on the unsupervised learning tasks to demonstrate generalizability of the transfer-learning and interpretability of the cell and gene embeddings. If you think that kNN classification is an important application, we can perform more in-depth analyses in the next revision.

- Please cite scArches <https://www.biorxiv.org/content/10.1101/2020.07.16.205997v1>.

#23 Our response:

Thank you for your suggestion! We have added the citation.

Gene set enrichments:

- It is important to plot top N pathways and enrichment p-values given in Table S4,S5,S6. The readers should know how relevant top pathways are.

#24 Our response:

We agree with the reviewer that visualizing the enrichment p-values for top N pathways would be helpful. Please also note that in the original submission, we used the top 30 genes per topic to test for pathway enrichment using hypergeometric tests. In this revision, we replace it with Gene Set Enrichment Analysis (GSEA) to test whether the weighted rank of the genes by the

topic score per topic is significantly enriched for a gene set, thereby omitting the need to arbitrarily choose the top genes per topic. This was done by calculating a running sum of enrichment scores by going down the gene list that is sorted in the decreasing order by a given topic distribution. Please see **Pathway enrichment analysis** under **Methods** for details.

To directly address this comment, we first tried to plot the BH-corrected q-values of top 5 pathways (**Figure R24**). However, as GSEA estimates the significance of enrichment by permutation test, the lowest q-value that can be achieved is bounded by the number of permutations, and therefore we believe this plot conveys very little information. Instead, we decided to display the following:

1. The negative log₁₀ of q-values for all pathway-topic associations in the form of a Manhattan plot (inspired by the GWAS Manhattan plot) to have an overall view of the significant gene sets (**Figure R25-1**; also **Figure 6** in the main text)
2. The list of all pathway names that have enrichment q-values below a certain threshold (e.g., 0.01 for MDD) and the number of association topics (**Supplementary Table S18**);
3. A shorter version of 2), only showing relevant pathways and the number of topics or dimensions that are enriched in these pathways in comparison with the scVI-LD (**Table R24-1,2,3**; **Supplementary Tables S12, S13, S14**).

Pathway Name	# scETM Topics	# scVI-LD latent dimensions
REACTOME_INSULIN_PROCESSING	15	2
REACTOME_INSULIN_RECEPTOR_RECYCLING	6	0
PID_INSULIN_GLUCCOSE_PATHWAY	4	0
WP_PANCREATIC_ADENOCARCINOMA_PATHWAY	2	0
KEGG_PANCREATIC_CANCER	2	0
PID_INSULIN_PATHWAY	2	0
SIG_INSULIN_RECEPTOR_PATHWAY_IN_CARDIAC_MYOCYTES	1	0
REACTOME_REGULATION_OF_GENE_EXPRESSION_IN_LATE_STAGE_BRANCHING_MORPHOGENESIS_PANCREATIC_BUD_PRECURSOR_CELLS	1	0
REACTOME_REGULATION_OF_INSULIN_SECRETION	1	1
REACTOME_REGULATION_OF_INSULIN_LIKE_GROWTH_FACTOR_IGF_TRANSPORT_AND_UPTAKE_BY_INSULIN_LIKE_GROWTH_FACTOR_BINDING_PROTEINS_IGFBPS	1	3
WP_INSULIN_SIGNALING	1	0
BIOCARTA_INSULIN_PATHWAY	1	0
REACTOME_INSULIN_RECEPTOR_SIGNALLING_CASCADE	0	1
REACTOME_SYNTHESIS_SECRETION_AND_INACTIVATION_OF_GLUCCOSE_DEPENDENT_INSULINOTROPIC_POLYPEPTIDE_GIP	0	1
REACTOME_SIGNALING_BY_TYPE_1_INSULIN_LIKE_GROWTH_FACTOR_1_RECEPTOR_IGF1R	0	1
WP_FACTORS_AND_PATHWAYS_AFFECTING_INSULINLIKE_GROWTH_FACTOR_IGF1AKT_SIGNALING	0	1
Total number of pathways	12	7

Table R24-1 Pathway enrichment statistics for Human Pancreas dataset. Pathways whose name includes the keywords "insulin" or "pancreatic" are shown. For each pathway, the number of topics with significant enrichment (BH-corrected q-value < 0.01) are counted. Both scETM and scVI-LD use latent dimensions of 100 for fair comparison. The same table is displayed in **Supplementary Table S12**.

Pathway Name	# scETM Topics	# scVI-LD latent dimensions
REACTOME_AMYLOID_FIBER_FORMATION	1	2
KEGG_ALZHEIMERS_DISEASE	1	2
REACTOME_DEREGULATED_CDK5_TRIGGERS_MULTIPLE_NEURODEGENERATIVE_PATHWAYS_IN_ALZHEIMER_S_DISEASE_MODELS	1	0
Total number of pathways	3	2

Table R24-2 Pathway enrichment statistics for AD dataset. Pathways whose name includes the keywords "amyloid" or "alzheimer" are shown. For each pathway, the number of topics with

significant enrichment (BH-corrected q-value < 0.01) are counted. Both scETM and scVI-LD use latent dimensions of 100 for fair comparison. The same table is displayed in **Supplementary Table S13**.

Pathway Name	# scETM Topics	# scVI-LD latent dimensions
SUBSTANCE/DRUG INDUCED DEPRESSIVE DISORDER	2	2
Total number of pathways	1	1

Table R24-3 Pathway enrichment statistics for MDD dataset. Pathways whose name includes the keyword "depressive" are shown. For each pathway, the number of topics with significant enrichment (BH-corrected q-value < 0.01) are counted. Both scETM and scVI-LD use latent dimensions of 100 for fair comparison. The same table is displayed in **Supplementary Table S14**.

- Repeating model fits and gene enrichments only with ~20k protein-coding genes on standard chromosomes might improve interpretability considerably. Right now, there is a lot of AC, AL and LINC genes in Fig S1.

#25 Our response:

Thanks for this suggestion. First of all, as mentioned above, we have switched to GSEA, which eliminates arbitrarily choosing the top genes compared to the hypergeometric tests. Therefore, we used GSEA throughout our analysis. Based on your comment, we performed the model training and GSEA with only coding genes for the MDD dataset ("MDD - coding genes only"). Compared to the embedding from the MDD dataset using all of the genes, we observe less significant MDD-related gene-set enrichment (q-value = 0.001 compared to 1e-5 when using all of the genes, **Figure R25-1**). Despite the removal of the AC, AL and LINC genes, we do not find significantly more known marker genes among the top 10 genes per topic (**Figure R25-2**). Because the regulatory interplays among non-coding RNAs, RNA binding proteins, and the protein-coding genes are indicative of complex diseases (Geauer et al., Nat Rev Genet 2021), it is possible that including these noncoding genes can in fact improve the interpretability of the topics. Therefore, we primarily focus our analysis on the embeddings obtained from all of the genes in the main text.

Reference:

Gebauer, F., Schwarzl, T., Valcárcel, J. & Hentze, M. W. RNA-binding proteins in human genetic disease. Nat Rev Genet 22, 185–198 (2021).

MDD - all genes

MDD - coding genes only

Figure R25-1. GSEA of a MDD-related gene set discovered among the scETM topics. scETM was trained on the MDD snRNA-seq dataset with 100 topics using either all of the genes (including noncoding genes such as lincRNA; left panel) or only the protein-coding genes (right panel). (a) Manhattan plot of the GSEA with x-axis as the gene sets and y-axis as the $-\log$ q-values. One of the relevant gene set substance-induced depressive disorder is annotated on the plot. (b) GSEA leading edge plot for substance-induced depressive disorder. We displayed the gene ranked by topic 13 from scETM trained on *MDD - all genes* and topic 49 from scETM trained on the *MDD - coding genes only*. (c) UMAP visualization of the topic embeddings and gene embeddings. Genes from the substance-induced depressive disorder and the enriched topics are annotated on the plot. The same figures are in **Supplementary Figures S12, S13**.

Figure R25-2. Gene topics heatmap of top 10 genes in cell-type differential topics, ordered based on topic intensity. Top panel: MDD topics derived using all of the genes; bottom panel: MDD topics derived using only the protein-coding genes. Known cell-type markers genes are highlighted. Condition DE topic is a DE topic for the MDD subjects compared against the control subjects; Cell-type DE topic is a DE topic for the cell type compared against the rest of the cell types. The same figures are displayed in **Supplementary Figures S18 and S19**.

- Figure 4 Panel A: highlight the genes that are involved in the pathway written underneath the heatmap

#26 Our response:

Thanks for the suggestion. We highlighted the genes in two different ways as shown in the GSEA leading edge plot (**Figure R25-1b, Figure 5**) and the heatmap (**Figure R25-2, Figure 6**). Specifically, for the pathway enrichment, we included standard GSEA leading edge plots in which the “leading edge” subset represents the core group of genes that accounts for the gene set's enrichment signal. For the cell-type DE topics (shown in heatmap), we highlighted the genes that are known as cell-type marker genes by comparing them with a marker gene database called PanglaoDB (<https://panglaodb.se/>).

- Although gene and topic embeddings are an extremely cool feature of the method, they are not shown at all. I suggest plotting gene and topic embeddings for some topics as in Dieng et al paper.

#27 Our response:

Agreed! **Figure R27c (Figure 5c** in the main text) illustrates the gene embeddings and topic embeddings learned from the HP datasets and projected onto a two-dimensional plane by the UMAP algorithm. Given that the dataset was derived from human pancreas, genes from **Insulin Processing from Reactome** and the enriched topic 54 by the GSEA are annotated on the figure.

We also examined the scETM-learned topic embeddings and gene embeddings for Alzheimer's Disease (AD) gene set from KEGG based on the AD scRNA-seq data (**Supplementary Figure S11**) as well as topic enriched for the substance-induced depressive disorder gene set based on the MDD dataset (**Figure R25-1; Supplementary Figure S12, S13**). In these two datasets, although the localization of the genes and enriched topic are not as prominent as the HP results, we observe an overall consistent pattern between the GSEA and the scETM-learned embeddings.

- It is not clear why the color scale in Panel C has so extreme values, compared to the [-4,6] range in Panel B. Setting the maximum values for zscores makes it hard to see smaller differences. Could the authors comment on that?

#28 Our response:

Sorry about the confusion. In Figure 4 of the initial submission, the colors in Panel B show the topic activities (i.e., the unnormalized topic values) and the colors in Panel C show the z-score from one-sided t-test using the topic activities.

We agree with the reviewer that the z-scores are very extreme, which implies that the topic activities may not meet the normality assumption in the t-test. Therefore, we switched to permutation tests for the differential analysis in this revision.

For topic k and cell label j (i.e., cell type or disease condition), we first calculated the difference of the average topic activities between the cells with label j and the cells without label j . For each permutation trial, we randomly shuffled the label assignments among cells and recalculated the difference of average topic activities from the resulting permutation. The empirical p-value was calculated as $(N'+1)/(N+1)$, where N' is the number of permutation trials in which the difference is greater than the observed difference, and N is the total number of permutation trials. To account for multiple hypotheses, we applied Bonferroni correction by multiplying the p-value by the product of the topic number and the number of labels. We performed $N=100,000$ permutations.

The above description was added to **Methods section Differential analysis of topic expression. Figure 5c** has been updated accordingly for AD (**Figure R28; Figure 6**) and MDD (**Supplementary Figure S18, S19**).

p-scETM:

- Adding a conceptual plot in Figure 5 explaining how rho matrix is prepared and how the overall model looks like would help the readers.

#29 Our response:

Good idea. We added this in **Figure 1a (Figure R5a)** and **Figure 7a (Figure R30a)** below). We also referred the readers to **Methods section Incorporation of pathway knowledge into the gene embeddings**.

- As a popular gene set resource, please consider providing the GO version of p-scETM.

#30 Our response:

We have repeated model fits with Gene Ontology - Biological Processes (GO-BP) dataset (**Figure R30; Figure 7** in the main text). Similar to what we reported for pathDIP version of p-scETM, the GO version of p-scETM is also able to learn topics whose top pathways are relevant to insulin signaling and pancreatic functions in HP, pathways relevant to amyloid protein regulation in AD, and neuronal and G-protein related pathways for MDD.

a. Pathway-informed scETM (p-scETM)

b. topic embeddings from the four scRNA-seq datasets

Figure R30 (Figure 7 in main text). Pathway-topics embeddings learned by the pathway-informed scETM (p-scETM). **(a)** p-scETM overview. Pathways information as pathways-by-genes are provided as the gene embedding in the linear decoder. The learned

topic embedding α is the direct association between the topics and pathways. **(b)** The pathway-topics heatmap of top 5 pathways in selected topics. Here the pathways are the Gene Ontology - Biological Processes terms. For the HP dataset, GO-BP terms whose names include the keywords "insulin" or "pancreatic" were highlighted. For the AD dataset, GO-BP terms whose names include the keywords "amyloid" or "alzheimer" were highlighted. For MDD - all genes and MDD - coding genes only, GO-BP terms whose names include the keywords "neuron" or "G-protein" were highlighted.

Code:

[- Please cite PyTorch \(https://github.com/pytorch/pytorch/blob/master/CITATION\) and give credit to Adji Dieng for the implementation if your implementation is not from scratch.](https://github.com/pytorch/pytorch/blob/master/CITATION)

#31 Our response:

Thank you. We cited PyTorch. Our initial implementation was based on Dieng's code. However, later we rewrote the code for optimal scalability. We have added the citation and acknowledgements in the text in section **scETM implementation details** under **Methods**.

- Unfortunately the code is not available, please make it available.

#32 Our response:

Thanks for the suggestion! We have released our code to the public at <https://github.com/hui2000ji/scETM>. It is also packaged and released on PyPI so one can easily install the package by entering `pip install scETM` in the terminal.

The package is integrated with scanpy (details in **Response #33 to Reviewer 1**) and tensorboard. Users can view the cell, gene and topic embeddings interactively via tensorboard. Additionally, our GitHub repo contains a detailed tutorial of our model (including all the variants) and the scripts that we used to run other methods.

- Especially for the scrna-seq tools implemented in Python, integration with scanpy, as the most popular framework in the field, is critical. It would be great to have a function e.g. `scetm.tl.infer(adata)` which stores topic and gene embeddings as well as the loadings in `.obsm` and `.varm` fields of an `AnnData` object.

#33 Our response:

Thank you for your suggestion! Our implementation is inherently compatible with scanpy, just as you described. For example, one can easily train a scETM as follows:

```
from scETM import scETM, UnsupervisedTrainer
model = scETM(adata.n_vars, adata.obs.batch_indices.nunique())
trainer = UnsupervisedTrainer(model, adata)
```

```
trainer.train(save_model_ckpt = False)
model.get_all_embeddings_and_nll(adata)
```

The above code snippet will instantiate an scETM model, train the model, infer the unnormalized cell topics mixture of adata and store them in `adata.obsm['delta']`. We can also access the gene and topic embeddings via `adata.varm['rho']` and `adata.uns['alpha']`. We added this to **Methods** section **scETM software**.

- Please describe which package is used for ARI and NMI and cite if necessary.

#34 Our response:

We calculate ARI, NMI and ASW using the Python library scikit-learn. We added the citation of the used packages in the **Methods section Clustering performance benchmark of the existing methods**.

- Consider using the NB distribution when scVI is applied to UMI datasets, since UMI data is not zero inflated, see Suppl. Figure 1 in the DCA paper.

#35 Our response:

We performed experiments on the NB and ZINB variant of scVI on Mouse Pancreas and Human Pancreas. The results of using NB and ZINB likelihood are actually quite similar as shown in the table below. The ZINB in scVI generalizes NB because the zero-inflation mixing proportion is also modeled by a neural network. In other words, even though the UMI data is not zero inflated, the end-to-end deep learning framework can learn a negligible zero-inflation mixing proportion so that the final distribution is very close to NB. Because the Human Pancreas dataset contains both UMI and non-UMI batches, it may be better to let the model decide how to best fit the data.

Mouse Pancreas

	ARI	NMI	kBET
scVI - NB (100)	0.8792	0.8442	0.4167
scVI - ZINB (100)	0.8710	0.827	0.4883

Human Pancreas

	ARI	NMI	kBET
scVI - NB (100)	0.8399	0.8451	0.0400
scVI - ZINB (100)	0.8507	0.8474	0.0448

Conceptual questions:

- How are “stop words” (e.g. MALAT1) handled? Do you see ubiquitously expressed genes dominating all topics?

#36 Our response:

Thanks for the question. When using the original ETM, we observe that ubiquitously expressed genes such as *MALAT1* tend to appear among the top genes in several topics. Our scETM corrects the background gene expressions by the gene-dependent bias vector λ , which may differ from batch to batch or the same across all samples when sample-batch is of lesser concern. As a result, the ubiquitously expressed genes do not dominate all topics from scETM. We added the above point into the **Discussion** section.

- How is the variation in library sizes (see the l_n variable in scVI paper) handled?

#37 Our response:

As input to the encoder to infer cell topic mixture, we use sum-normalized counts by dividing them by the total counts in that cell such that each cell would have a total count of 1. This effectively addresses different library sizes in each cell and different sequencing depth in the scRNA-seq protocols. Although we used the sum-normalized counts as input to the encoder, we model the categorical likelihood (i.e., the negative reconstruction loss) using the raw counts. Prompted by your comments, we also tried using the sum-normalized count in the loss function, which prevents batches with large total counts from receiving significantly larger penalties. As shown in the table below, we found similar performances in Human Pancreas and Tabula Muris comparing using unnormalized counts and the normalized counts in the reconstruction loss.

	Human Pancreas		Tabula Muris	
	ARI	kBET	ARI	kBET
Unnormalized loss	0.9409	0.055	0.7771	0.052
Normalized loss	0.9431	0.081	0.7606	0.079

- Is the encoder input log transformed and/or TP10K normalized?

#38 Our response:

As explained in our Response #37 above, our encoder input is just counts divided by total counts, without any further transformations.

Minor:

- I suggest supporting this statement with a citation or remove: "Variational autoencoders (VAE) [26] is an efficient probabilistic framework known to better account for noise compared to conventional autoencoders."

#39 Our response:

Agreed. We have modified the statement in the **Background** section to "Variational autoencoders (VAE) (Kingma & Welling, 2013) is an efficient Bayesian framework for approximating intractable posterior distribution using proposed distribution parameterized by neural networks. Several recent studies have tailored the original VAE framework towards modeling single-cell data."

- The authors use read count matrix and read counts throughout the manuscript however some datasets use UMI counts and not read counts. I suggest removing "reads" and just using "counts" to imply that it can be read or UMI counts.

#40 Our response:

Thanks for your suggestion. We have revised the text accordingly.

Reviewer #2 (Expertise: Deep learning with an application to scSeq data analysis):

In this work, the authors propose single-cell Embedded Topic Model (scETM) to obtain interpretable embeddings of single cell data matrix. They mention that scETM is more scalable, obtain high interpretability and is able to transfer knowledge across datasets. Also they show how pathway information could be used in the learning step.

Strengths of the work:

- i. The paper is well-written and easy to follow. The authors provide conditions in which they ran the baselines, provided significance values in hypothesis testing scenarios and compared multiple state-of-the-art methods on their datasets.*
- ii. Transferring the trained model on Human genes to mouse genes still obtained good performance (while competing methods suffer) is an interesting finding.*
- iii. Fixing the pathway matrix and then obtaining interpretable topics is also an interesting finding.*

#1 Our response:

Thank you very much for the positive comments.

Weaknesses:

i. Lack of novelty: Deep learning methods for clustering single cells is not new in the literature. In fact, the proposed approach is very similar in spirit to scVI where the authors also used a VAE. scVI-LD also uses a linear decoder for interpretability just as proposed here. The matrix factorization approach to single cell clustering and dimension reduction has been proposed in prior published work [1]. Considering all these, I find the contribution of the current work limited.

#2 Our response:

Thanks for pointing this out. It is true that there have been many non-deep-learning and deep-learning methods developed recently for clustering single-cell RNA-seq data. This does not only highlight the importance of the problem but also the unsolved challenges. Every existing method has its own strengths and weaknesses.

As you kindly pointed out, earlier non-deep-learning methods such as UNCURL [Mukherjee et al., Bioinformatics 2018] and LIGER [Welch et al., Cell 2019] used non-negative matrix factorization (NMF) to solve a linear problem. The cell loadings are then used for clustering cells. While UNCURL works only with one dataset, LIGER can integrate multiple datasets linking cells from different conditions by a common set of latent factors also known as metagenes. As observed in much earlier study by Kim and Tidor (Genome Research 2003) in the application of bulk microarray gene expression data, the strength of NMF method is the high interpretability, which is the reason it remains popular despite its restricted linear assumption and scalability issues on massive-scale genomic data. Same goes to the topic models that become popular in computational genomics because of its high interpretability in the topic probabilities years after it was introduced in 2003 by Blei on text mining. For example, cisTopic published in Nature

Methods is a direct implementation of collapsed Gibb sampling for LDA. The novelty in these works lies in their applications and biological findings, which is reflected in our experiments.

We would also like to articulate our technical contribution. Indeed, many state-of-the-art methods used deep learning frameworks such as scVI to learn more flexible non-linear embedding in a much more efficient way via stochastic gradients than NMF. Because of the distributed representation of the neural network, the trade-off of these methods is the interpretability when compared to NMF and topic models (as we described in the **Background** section). To address this, our scETM uses linear decoder to improve interpretability and Gaussian-VAE to perform inference. Although these two concepts are also used in scVI-LD, the two models have very different data generative processes. In particular, scVI-LD assumes that for each cell d :

1. $z_d \sim \text{Normal}(0, I)$
2. $s_d \sim \text{Logistic-Normal}(s_{\mu}, s_{\sigma})$
3. $u_d = z_d W$ (where W is the $K \times G$ linear decoding weights for K factors and G genes)
4. for $g = 1, \dots, G$:
 - a. $v_{d,g} \sim \text{Gamma}(\theta_{g, d}, \mu_g)$
 - b. $y_{d,g} \sim \text{Poisson}(v_{d,g} * s_d)$

In contrast, our scETM follows a much simpler topic-model-style generative process with a *tri-factorization design* to incorporate topic embedding (α) and gene embedding (ρ) as well as the batch bias (λ) in a linear fashion:

1. $\theta_d \sim \text{Logistic-Normal}(0, I)$
2. For $g = 1, \dots, G$:
 - a. $r_g = \text{softmax}(\theta * \alpha * \rho_g + \lambda_g)$ (this step is the deterministic linear decoding step)
 - b. $y_{d,g} \sim \text{Categorical}(r_{d,g})$

As we demonstrated in our experiments, this model formulation allows us to accomplish the following tasks more successfully compared to scVI-LD and scVI:

- **Data integration (batch correction via linear decoder)**
 - Learning a common set of cellular programs in the topics by integrating multiple batches by simultaneously learning the linear batch effects
- **Transfer learning application**
 - Better performance in terms of ARI because of the joint learning of gene embedding and topic embedding
- **Learning gene and topic embeddings:**
 - The novel introduction of the topic and gene embeddings in modeling scRNA-seq allows:
 - more accurate cross-tissue and cross-species transfer learning (please see our **Response #20** and **#21 to Reviewer 1**)

- analyze shared and unique cellular programs implicated in each topic by learning connections between topics and genes via common embeddings (please see **Response #27 to Reviewer 1**)
- better interpretability than scVI-LD in terms of enrichment for known biological pathways. For details, please see our **Response #4** to your comment.
- **Incorporating known gene sets via gene embedding:**
 - scETM allows the use of pre-trained gene embedding rho by incorporating existing ontology knowledge. In particular, we demonstrated p-scETM that uses MSigDb gene-set matrix or Gene Ontology terms as fixed gene embedding to further improve interpretability (please **Response #30 to Reviewer 1** and the revised **Section Pathway-informed scETM topics** in **Results**).
- **High scalability due to efficient deep learning implementation:**
 - Software usage side we also manage to improve computational efficiency, which is the key behind many successful deep learning methods. For details, please our **Response #7** to your comment.
- **Better interpretability with more enriched pathways based on GSEA:**
 - Please see our **Response #3** to your comment.

Lastly, we would like to re-emphasize the importance and the thoroughness of the above analyses carried out in our current study. Many of them (e.g., zero-shot transfer learning, GSEA enrichment of topics, visualizing topic embeddings and gene embeddings, analyzing topic embeddings of known pathway or gene sets, etc) were not demonstrated in scVI, scVI-LD, or (to the best of our knowledge) any existing paper. Inspired by this comment, we have revised the **Background** and **Discussion** to emphasize the above points in highlighting the novelties of our approach.

ii. The baseline/competing methods are run with their default settings, while the proposed scETM could leverage different hyperparameters. While the authors do show a robustness to hyperparameters table, it would be nice to compare baselines on datasets which they used in their publication (Example - strongly recommend showing results of scETM on either of CORTEX or HEMATO datasets made publicly available by scVI authors). This would lead to a more direct comparison. Also, it would be nice to see what scVI clustering metrics look like for latent dimension of 100 (equivalent to 100 topics used by scETM). The current latent dimension of 10 seem too small for scVI.

#3 Our response:

Thank you for this suggestion. We first would like to emphasize that

1. scETM is not very sensitive to hyperparameter changes (**Supplementary Table 2**).
2. We use the same set of hyperparameters throughout our benchmark in Table 1. For example, although we found that using a larger number of topics for complex atlas datasets like the Tabula Muris (FACS) dataset yields better results, we stick to the default number of topics.
3. We *are* using datasets that were used by competing methods in their publications, e.g., the Human Pancreas dataset is used in the Seurat and Harmony paper.

We have experimented with our method and others on the Cortex dataset. This dataset does not have different batches and all of the methods achieve reasonably high ARI as shown in the table below. Since our focus in the study is to demonstrate integrative methods on modeling multiple datasets with batch correction, we decided not to use this dataset in the **Data Integration** section.

Method	ARI on the Cortex dataset
Harmony	0.836
Scanorama	0.767
Seurat Integrated	0.838
scVAE-GM	0.858
scVI	0.820
LIGER	0.852
scVI-LD	0.806
scETM	0.844

Therefore, we focused in comparing the performance on datasets with more confounding batch effects such as HP or large-scale integration tasks such as TM (FACS), scalability, transfer learning capabilities (See **Response #21 to Reviewer 1**), interpretability to comprehensively reveal the strengths and weaknesses of each model.

We also compared the performance of scVI(-LD) with 10 and 100 latents on five scRNA-seq datasets. We found that in most cases, scVI and scVI-LD with 10 latents perform better than their 100-latent variants, justifying the hyperparameter choices made by the scVI(-LD) authors. We added this in the **Methods scVI/scVI-LD description**.

ARI:

	MP	HP	MDD	AD	TM
scVI-LD (10)	0.8753	0.6563	0.6401	0.9889	0.6076
scVI-LD (100)	0.4454	0.4431	0.6145	0.9848	0.6705
scVI (10)	0.9325	0.7590	0.5411	0.9915	0.6699
scVI (100)	0.8215	0.7051	0.5813	0.9921	0.6284

kBET:

	MP	HP	MDD	AD	TM
scVI-LD (10)	0.1483	0.0335	0.2759	0.3967	0.0687
scVI-LD (100)	0.4601	0.0045	0.2745	0.3570	0.0331
scVI (10)	0.5159	0.1403	0.3132	0.4338	0.0557
scVI (100)	0.6856	0.0876	0.3402	0.4371	0.0301

iii. It is not clear how scETM is better with regards to interpretability for the range of experiments conducted. There are no metrics for interpretability. What are the interpretable clusters obtained from an approach such as scVI-LD on AD or MDD datasets?

#4 Our response:

We agree with the Reviewer that we were mainly using pathway enrichment to qualitatively assess the interpretability of the topics. To address this comment, we have conducted gene set enrichment analysis (GSEA) for the topics inferred by scETM and the gene loadings learned by scVI-LD using the Human Pancreas (HP), Alzheimer's Disease (AD) and Major Depressive Disorder (MDD) scRNA-seq or snRNA-seq datasets. Please see our Response #5 below for the details of GSEA.

We then compared the interpretability of the scETM and scVI by the number of significantly enriched pathways that are biologically relevant. For example, in HP, we used the pathways related to insulin signaling and/or pancreatic functions as the true positives. For fair comparison, we trained and evaluated scVI-LD with 100 latent dimensions and selected the dimension that is the most significantly enriched for the true positive pathways. As shown by the **Supplementary Tables S12,S13,S14** (also **Table R24-1,2,3** in our **Response #24** to Reviewer 1), scVI-LD gene loadings show fewer biologically relevant pathway enrichments than scETM in HP, AD and MDD.

We also examined the level of significance for the enrichment found in the biologically relevant pathways. We find that scETM's enrichment results are 2-3 orders more significant than those of scVI-LD (**Figure R4; Supplementary Figure S14**). We reason that the improvement over scVI-LD is attributable to the joint learning of the gene embedding and topic embedding.

a HP, scETM, FDR qval<0.0001

b HP, scVI-LD, FDR qval=0.00084

c AD, scETM, FDR qval<0.0001

d AD, scVI-LD, FDR qval=0.0001

e MDD, scETM, FDR qval<0.0001

f MDD, scVI-LD, FDR qval=0.0033

Figure R4. GSEA leading edge plots for scETM (left column) and scVI-LD (right column) on Human Pancreas (HP), Alzheimer's Disease (AD), and Major Depressive Disorder (MDD). Due to the disease relevance, we showed Reactome Insulin Processing, KEGG Alzheimer's Disease, and PsyGeNet substance/drug induced depressive disorder for HP, AD, and MDD, respectively.

iv. It is not clear to me what the false discovery of significant genes is in topics associated with cell types. For example, Page 6 Last paragraph "In AD, the top 30 genes from topic 75 are highly enriched in amyloid fiber formation". Are all the 30 genes related to amyloid fiber formation in the biology literature? Or are there any false positives? Similarly for the statement, "In MDD, topic 7 is enriched for neurodegenerative diseases such as Parkinson's , ...". In this line, there is not even a mention of how many top genes in the topic led to this claim.

#5 Our response:

Thanks for pointing this out. We do not aim to perform hypothesis tests at the level of individual genes, which is known to have limited statistical power due to the large number of multiple testings, subtle differences between cell types or conditions, and small sample size at the subject-level conditions (e.g., 17 MDD subjects vs 17 controls) [Nagy et al., 2020]. Instead, we aimed to perform pathway enrichment tests *at the level of scETM-learned topics*, which we interpret as the *de novo* cellular programs cognate with the underlying biology of the scRNA-seq data. We admit that the top 30 genes per topic were arbitrarily chosen out of *ad hoc* empirical analysis. In this revision, we switched to Gene Set Enrichment Analysis (GSEA), which eliminates thresholding genes and uses all of the genes in a weighted rank-sum test [Subramanian et al., 2005].

Briefly, the GSEA calculates a running sum of enrichment scores by going down the gene list that is sorted in the decreasing order by their association statistic with a phenotype. In our context, we treated the gene scores under each topic from the genes-by-topics beta matrix as the association statistic. The final ES for a gene set S is the maximum difference between $P_{hit}(S,i)$ and $P_{miss}(S,i)$, where $P_{hit}(S,i)$ is the average topic score of genes in S upto gene index i in the sorted list, and $P_{miss}(S,i)$ is the average topic score of genes not in S up to gene index i in the sorted list. The FDR for each gene set is computed by permutation tests via randomly shuffling the gene symbols on the sorted list (while keeping the gene topic scores in the decreasing order) 1000 times to compute the null distribution of the ES for each gene set and each topic.

We identified many pathways and gene sets below Benjamini-Hochberg corrected q-value threshold of 0.05 (**Supplementary Table S18**). Examples of relevant pathways to Human Pancreas (HP), Alzheimer's Disease (AD), and Major Depressive Disorder (MDD) are listed in **Table R24-1,2,3 (Supplementary Table S12, S13, S14)** (please see our **Response 24** to Reviewer 1) -- in comparison with the gene loadings learned by scVI-LD. Three examples of the enriched pathways are also illustrated in the leading edge plots in **Figure R4**.

We revised this in **Pathway enrichment analysis** under **Methods**.

References:

- Nagy, C., Maitra, M., Tanti, A., Suderman, M., Th roux, J.-F., Davoli, M. A., Perlman, K., Yerko, V., Wang, Y.-C., Tripathy, S. J., Pavlidis, P., Mechawar, N., Ragoussis, J., &

Turecki, G. (2020). Single-nucleus transcriptomics of the prefrontal cortex in major depressive disorder implicates oligodendrocyte precursor cells and excitatory neurons. Nature Publishing Group, 23(6), 1–18. <https://doi.org/10.1038/s41593-020-0621-y>

- Subramanian, A., Tamayo, P., Mootha, V. K., Mukherjee, S., Ebert, B. L., Gillette, M. A., Paulovich, A., Pomeroy, S. L., Golub, T. R., Lander, E. S., & Mesirov, J. P. (2005). Gene set enrichment analysis: a knowledge-based approach for interpreting genome-wide expression profiles. Proceedings of the National Academy of Sciences of the United States of America, 102(43), 15545–15550. <https://doi.org/10.1073/pnas.0506580102>

Additional comments:

1. Page 5 Last paragraph "We trained an scETM model on TM-FACS and .. evaluated it using the MP data, which only contains mouse pancreatic islet cells". Does the TM-FACS dataset not contain any mouse pancreatic islet cells?

#6 Our response:

Yes, the TM (FACS) reference dataset does contain pancreatic cells. Our main point was that: scETM can transfer knowledge obtained from atlas data to help distinguish cell types with small numbers of cells in a target dataset. However, your comments also prompted us to explore another analysis: training scETM on TM (FACS) *with pancreas removed* (TM (FACS) w/o Pancreas) and then apply to the MP dataset. The results are shown below:

	MP -> MP	TM-FACS -> MP	TM (FACS) w/o Pancreas -> MP
ARI	0.9457 ± 0.0086	0.9409 ± 0.0033	0.8072 ± 0.0449
kBET	0.2697 ± 0.0199	0.3388 ± 0.0342	0.4772 ± 0.0166
RCD(B-cell, macrophage)	0.2207 ± 0.1250	0.7214 ± 0.0173	0.6686 ± 0.0733
RCD(macrophage, B-cell)	0.6898 ± 0.0303	0.8620 ± 0.0050	0.8368 ± 0.0232
RCD(T-cell, macrophage)	0.3561 ± 0.1785	0.7063 ± 0.0308	0.7002 ± 0.0562
RCD(macrophage, T-cell)	0.7635 ± 0.0136	0.8654 ± 0.0098	0.8413 ± 0.0163
RCD(B-cell, T-cell)	0.3243 ± 0.0231	0.4612 ± 0.0390	0.4307 ± 0.0205
RCD(T-cell, B-cell)	0.2814 ± 0.0389	0.4178 ± 0.0513	0.4698 ± 0.0226

Here $RCD(A, B) = (d_{AA} - d_{AB}) / \max(d_{AA}, d_{AB})$ is a “relative cluster distance” metric inspired by the average silhouette width, where d_{AA} is the mean intra-cluster cosine distance for cluster A, and d_{AB} is the mean inter-cluster cosine distance between clusters A and B. The more separated the two clusters, the higher this value. We observe that the TM (FACS)

reference set successfully helps scETM discriminate B-cells, T-cells and macrophages from each other regardless of the presence or absence of the pancreas islet cells from TM (FACS).

We also extended the transfer learning experiment to more dataset pairs, demonstrating scETM's superior transferability across technology, tissues and species. For instance, we found that scETM trained on MP alone can in fact separate cells of many unknown cell types from the much larger TM (FACS) dataset. Please see our **Response #20 and #21 to Reviewer 1** and the **Transfer learning** section for more details.

2. Page 5 Section Scalability. What aspect of the proposed model led to run-time gain is not clear. Stochastic variational inference, minibatch parameter update is applicable to other deep learning models such as scVI also. What aspect of scETM is unique for scalability, which cannot be applied for other deep learning methods?

#7 Our response:

Model-wise, scETM is similar to other deep learning models in terms of theoretical time and space complexity. We want to emphasize that implementation is also very important, **especially for deep learning models**. For example, scVAE is much slower and more memory consuming than scVI, while they are very similar VAE models. One of the main speedups provided by scETM comes from our implementation of a multithreaded data loader for minibatches, which does not need to be re-initialized every epoch as the standard PyTorch DataLoader. Also, compared to scVI and scVI-LD, the normalized counts in both the encoder input and the reconstruction loss used by scETM remove the need to infer the cell-specific library size variable, and the simpler categorical likelihood choice also helps reduce the computational time. We added a short description to the **Scalability** results section and a longer description in the **Discussion** section.

3. Page 8 Section Pathway-informed scETM Topics. "6 topics related to nutrient digestion and metabolism", "we found 8 topics with top pathways known as therapeutic targets for MDD treatment". How would you group together topics in a setting where ground truth pathway knowledge is not available? The number of topics for a group seems to vary. Not clear how to group together topics without labeled/annotated information.

#8 Our response:

Throughout our analysis, we assumed *a priori* that the topics were independent and only performed *post-hoc* enrichment analysis and quantified the number of topics that are commonly enriched for a relevant pathway or gene set. If we really wanted to group topics, we could have clustered topics based on their embeddings in a similar way as we cluster cells based on their topic mixture using Louvain or Leiden. However, we agree with the Reviewer that the lack of ground-truth pathway knowledge for each dataset makes it difficult to validate the scETM topics or the topic clusters.

Besides differential expression analysis and GSEA, we sought to achieve a more direct interpretation of the scETM topics, therefore we developed the pathway-informed scETM

(p-scETM) variant, where the gene embedding ρ (L by gene matrix) is fixed by a curated pathway database. In other words, instead of grouping topics together, we use the prior pathway information to guide topic interpretation. Your comment made us realize that the way we summarized the p-scETM results could be a little misleading to the readers. We revised the **Pathway-informed scETM topics** section accordingly.

Reference:

- Mantas, Ioannis, et al. "Update on GPCR-based targets for the development of novel antidepressants." *Molecular Psychiatry* (2021): 1-25.

Reproducibility:

I could not access the link to code (<https://github.com/li-lab-mcgill/scETM>) provided by the authors. Hence unable to ensure whether it is easy to reproduce the work.

#9 Our response:

We apologize for not releasing the code earlier. The code is now available at <https://github.com/hui2000ji/scETM>. It is also packaged and released to PyPI so one can install it easily with `pip install scETM`. The package is integrated with scanpy (details in **Response #33 to Reviewer 1**) and tensorboard. Users can view the cell, gene and topic embeddings interactively via tensorboard. Additionally, our GitHub repo contains a detailed tutorial of our model (including all the variants) and the scripts that we used to run other methods.

Minor typo:

Page 7 Last paragraph Line 6 "learn only the pathways by topics embedding" should be "learn only the topics by pathways embedding".

#10 Our response:

Thank you for your suggestion. We have corrected the text.

References:

[1] Mukherjee S, Zhang Y, Fan J, Seelig G, Kannan S. Scalable preprocessing for sparse scRNA-seq data exploiting prior knowledge. *Bioinformatics*. 2018 Jul 1;34(13):i124-32.

#11 Our response:

Thank you for this reference. We have cited this paper in the **Background** section when discussing NMF.

Reviewers' Comments:

Reviewer #1:

Remarks to the Author:

All points are addressed very well, no further comments. Thanks.

Reviewer #2:

Remarks to the Author:

The authors have adequately addressed all my past concerns. They have included detailed comparison on the common benchmark datasets, clarified the differentiating factors in their new datasets and shown results with latent dimensions 10 and 100 for scVI-LD.

- There is clarification now of the novelty and distinct steps from scVI-LD. I would encourage the authors to include these distinctions.

- There is more detailed result on the transfer learning, where the commonality (pancreatic genes) are missing.

- The code is now accessible with tutorials and easy to use pypi installation.

Overall, the revisions have increased the strength of the paper.

Point-by-Point Response to Reviewers

We would like to once again thank both Reviewers for the thoughtful and constructive comments.

Reviewer #1 (Remarks to the Author):

All points are addressed very well, no further comments. Thanks.

Our response:

We are glad to hear that. Thank you again for the detailed suggestions.

Reviewer #2 (Remarks to the Author):

The authors have adequately addressed all my past concerns. They have included detailed comparison on the common benchmark datasets, clarified the differentiating factors in their new datasets and shown results with latent dimensions 10 and 100 for scVI-LD.

- There is clarification now of the novelty and distinct steps from scVI-LD. I would encourage the authors to include these distinctions.

- There is more detailed result on the transfer learning, where the commonality (pancreatic genes) are missing.

- The code is now accessible with tutorials and easy to use pypi installation.

Overall, the revisions have increased the strength of the paper.

Our response:

Thank you very much for the positive comments and thank you again for the detailed suggestions in the last round.